# Methodology of Epidemic Risk Analysis in the Naval Military

**DOI:** 10.3390/ijerph22040572

**Published:** 2025-04-05

**Authors:** Laetitia Peultier-Celli, Alain Gérard, Franck Letourneur, Clara Inghels, Audrey Duclos, Philippe Perrin

**Affiliations:** 1Safety Environment and Crew Department, Naval Group, F-56100 Lorient, France; clara.inghels@ls2n.fr (C.I.);; 2Human Factors and UX design Department, Naval Group, F-83090 Ollioules, France; 3Research Unit DevAH—Development, Adaptation and Handicap, Faculty of Medicine, University of Lorraine, F-54500 Vandoeuvre-lès-Nancy, France; 4Infectious Diseases, Faculty of Medicine, University of Lorraine, F-54500 Vandoeuvre-lès-Nancy, France; 5Occupational Health Service, Naval Group, F-50100 Cherbourg-en-Cotentin, France

**Keywords:** pathogens characteristics, biological resilience, risk analysis

## Abstract

This review of the literature examines diseases and pathogen characteristics on military vessels, in order to improve the success of missions on a boat. Our aim is to understand the spread of disease, aiming to maximize biological resilience and hopefully eliminate outbreaks. Keyword research was conducted from various sources of information, including scientific publications, theses, public health organization websites, and clinical trials. A synthesis of bacterial, viral, fungal, and parasitosis characteristics was established, and a risk prioritization index was defined, based on contagiousness (basic reproduction number (R0)) and clinical severity. For instance, COVID-19 was assessed as moderately contagious, with critical severity, and Influenza A H1N1 as having a minor level of contagiousness with critical severity, resulting in a level two out of three risk prioritization index. This approach demonstrates that while diseases have numerous characteristics, a method for classifying them by isolating specific criteria and prioritizing them could be proposed. In conclusion, further work is needed to analyze onboard operator activities and develop simulation models related to pathogen characteristics.

## 1. Introduction

In the past, several outbreaks of infectious diseases have been reported on ships, emphasizing the challenges posed by enclosed, densely populated environments [1,2,3,4]. Ships, whether military or civilian, present unique conditions that can facilitate pathogen transmission due to limited space, shared facilities, and prolonged close-contact interactions. Among the epidemics identified were gastroenteritis on the cruise ships Aurora (2003) and Oasis of the Seas (2019) [5], pneumonia on the USS Boxer helicopter carrier (2007), the H1N1 influenza pandemic on the USS Theodore Roosevelt aircraft carrier (military) (2009) [6] and that on the Ocean Dream cruise ship (2009). The COVID-19 pandemic, which emerged in late 2019, further demonstrated the vulnerability of maritime settings. Large-scale outbreaks occurred on cruise ships such as the Diamond Princess and Grand Princess, and on Nile River cruises [7,8], as well as in the naval forces of several countries, including the French aircraft carrier Charles de Gaulle [9], the American aircraft carrier USS Theodore Roosevelt [10,11], and the destroyer USS Kidd [7]. The study by De Laval et al. [9] showed that a large proportion of the crew of the Charles de Gaulle was infected, highlighting the rapid spread of SARS-CoV-2 within a confined environment. Crowded conditions, shared ventilation, and frequent interactions among sailors facilitated the transmission of the virus. The majority of cases were symptomatic, but most cases were mild to moderate, with only a limited number of individuals requiring hospitalization. Nevertheless, despite attempts at isolation and social distancing on board, the virus spread before strict measures could be implemented, leading to the mission being cut short. On the USS Theodore Roosevelt, more than 1200 sailors were infected, representing approximately 25% of the crew. A portion of the crew was quarantined in Guam, and this had a major impact on the ship’s operational readiness [10]. This pandemic showed that an epidemiological crisis can have major effects on the availability of forces and their capacity to carry out missions. Indeed, in the confined environment of a ship with enclosed areas, there is inevitably close contact between crew members over an extended period of time, which may exacerbate disease spread. Consequently, ships can represent an ideal environment for the transmission of infections among crew members, and naval operations must integrate robust prevention and mitigation strategies to minimize the operational disruption caused by infectious diseases.

By detecting pathogens as early as possible, a ship’s crew can develop strong biological resilience, which will help control the impact of a potential epidemic threat. Biological resilience can be defined as “the ability to prevent, limit the effects of, and bounce back from biological disturbance linked to pathogenic elements of natural or terrorist origin. The aim is to enable military ships to recover function and operational efficiency as effectively as possible to a state equivalent to the prior disturbance” [12,13,14]. This biological resilience is based on a systemic approach to natural or terrorist-caused biological risks, establishing hypotheses to ensure the operational effectiveness of ships. This study is a scientific literature review aiming to define relevant diseases and the characteristics of associated pathogens. Indeed, there are numerous pathogens that vary in danger level, with several modes of transmission. It is necessary to identify those characteristics that favor transmission, along with detectability thresholds, clinical evaluation criteria, and acceptability thresholds.

The aim of this study was threefold: (1) to understand the intrinsic characteristics (standard values and severity thresholds) that favor the spread of pathogens, (2) to describe those diseases with a probability of occurrence in naval military settings and classify them according to a risk prioritization index that takes into account clinical severity and contagiousness, in order to target the most high-risk pathogens, and (3) to establish a roadmap for future studies to simulate pathogen propagation in naval forces. By integrating epidemiological modeling, risk assessment, and operational constraints, this research aims to enhance the resilience of military fleets and ensure their sustained operational effectiveness in the face of biological threats.

## 2. Literature Review

### 2.1. Search Strategy for the Characterization of Pathogens

The characteristics of the pathogens responsible for the various identified diseases were investigated. Pathogens can be classified into four categories: viruses, bacteria, fungi, and parasites.

In order to identify relevant studies on pathogen characteristics that influence the propagation of diseases, a keyword search was conducted across various knowledge bases, including PubMed, Google Scholar, Web of Knowledge, and ScienceDirect.

Different sources of information were used, such as scientific publications, theses, official public health organization websites, and clinical trials. The selection of articles was based on their relevance and the number of citations.

The research strategy was conducted in accordance with the preferred reporting items for systematic reviews and meta-analysis (PRISMA) guidelines. The PRISMA methodology was chosen for this study to ensure a rigorous and reproducible research process. This methodology provides a structured approach to systematically identifying, selecting, and analyzing relevant literature, which is essential for synthesizing high-quality evidence on pathogen transmission in naval environments. The use of PRISMA offers several advantages.

It minimizes selection bias by following a predefined inclusion and exclusion criteria framework, ensuring that only relevant and high-quality studies are considered.It enhances transparency by providing a detailed flow diagram of the study selection process, allowing for better reproducibility of the research.PRISMA improves the reliability of findings by promoting a comprehensive and structured literature review process, facilitating the identification of key trends and gaps in the existing knowledge.

The search used the following keywords: [pathogens] OR [bacteria OR viruses OR fungi OR parasitosis] AND [characteristics OR contagiousness OR spread OR transmission OR emission OR survival time]. The study selection was conducted based on several key criteria. Studies were selected within a defined publication timeframe (2010 to the present) to ensure the relevance of the findings. Only studies available in accessible languages, with a preference for English-language publications, were considered. Priority was given to studies focusing on pathogen transmission, epidemics, and biological resilience specifically in maritime contexts. Duplicate records and studies with limited applicability to the research scope were removed. Finally, to ensure scientific rigor, priority was given to peer-reviewed articles and high-quality studies. The flow chart is presented in Figure 1.

### 2.2. Characterization of Pathogens

#### 2.2.1. Pathogen Transmission

There are several modes of pathogen transmission, including airborne, direct contact, and vehicle transmission. On military ships, the greatest risk of contagiousness remains airborne transmission. Viruses are responsible for the majority of respiratory tract infections. Respiratory viruses can be transmitted via two primary modes: indirect contact (e.g., with a fomite, any object or clothing can carry infection), and air transmission (inhalation and direct deposition) [15,16]. Small droplets can quickly evaporate during the transition from the high relative humidity of the respiratory tract to the ambient air’s relative humidity. Azimi et al. (2020) [17] demonstrated that aerosol inhalation was probably the dominant contributor to COVID-19 transmission among the passengers of the Diamond Princess cruise, even under the conservative assumption of high ventilation rates and no air recirculation conditions.

#### 2.2.2. Contagiousness

Contagiousness can predict the spread of infectious diseases. Several factors influence contagiousness, for which the basic reproduction number (R0) acts as a means of quantifying contagiousness. R0 is defined as the average number of secondary infections produced by an infected individual in a susceptible host population [18]. Although R0 may vary, depending on ship-specific environmental factors (crew density, confined spaces, and ventilation systems that affect pathogen spread), control measures (vaccination and the presence of strict protocols), and contextual variabilities (population size, pre-existing immunity, crew fatigue, and limited access to medical care), it remains a factor to consider because R0 helps to evaluate the potential spread of an infectious disease within a specific environment, such as on a military vessel. Moreover, when identifying pathogens with a high R0, it is possible to prioritize infection prevention and control measures on board. Finally, R0 is a parameter used in epidemiological models to estimate the effectiveness of containment, ventilation, or vaccination strategies in enclosed environments. When the R0 value is less than 1, each infected individual, on average, transmits the pathogen to fewer than one other individual, leading to the pathogen’s eventual disappearance from the population. Conversely, when the R0 value is greater than 1, the number of cases increases on average over time, potentially leading to an epidemic [19]. Huang et al. (2021) [20] demonstrated in their study that a higher asymptomatic ratio leads to more infectious contacts.

R0 is a central parameter in the mathematical theory of epidemics. Assuming “panmixia” (i.e., randomness in a population), where the population is considered homogeneously susceptible, the basic reproduction number is the product of three parameters: the probability (β) of transmission of the virus during contact with the risk, the number of contacts (c) at risk, and the duration (d) of the generation interval between two infections (which are often equated with the length of the contagious period) [21].R0 = β × c × d(1)

R0 is not constant but is instead a random variable. It is not measured from epidemic curves but instead requires a back-calculation process based on the theory of epidemics and observation of the speed of propagation, particularly the time required for the number of cases to double. In the simple model mentioned above, the formula for estimating R0 is based on the estimation of doubling time (Td) and generation interval (d) [21].R0 = (d × ln(2) + Td)/Td(2)

For example, in the case of an epidemic with an observed doubling duration, where Td = 3 days and the generation interval d = 6 days, the R0 is 2.4.

Diseases transmitted through the air exhibit a range of R0 values that are not represented by a single average value but instead by thresholds that depend on many factors. For instance, tuberculosis has an R0 value ranging from 0.26 to 4.3, making it less transmissible than COVID-19, which has an R0 ranging from 1.4 to 8.9. Factors influencing airborne transmission include the viral load in respiratory particles of different sizes, the stability of the virus in aerosols, and the dose–response relationship for each virus.

Through experimental approaches, it is possible to estimate the number of infectious virus particles in contaminated fluids with the infectious dose 50 (ID50) criteria, i.e., the dose of virus required to infect 50% of exposed animals in animal studies. From dose–response studies in animals, we can establish a relationship between a dose of virus particles administered in units of PFU and the AID50 value, as was modeled for SARS-CoV in mice [22].

The unit of infectious dose used in infection risk prediction models is the quantum. A quantum is defined as the dose of airborne droplet nuclei required to cause infection in 63% of susceptible persons [23]. The quantum was originally a parameter calculated from the percentage of infected subjects in scenarios or aerosol challenge tests.

The quanta emission rate of the contaminated person (quanta.h^−1^) is estimated by considering certain physiological parameters (e.g., exhalation rate according to the level of vocal activity, speaking, singing, shouting, or physical activity) and virus-specific parameters (e.g., viral load in the exhaled air and number of infectious viral particles). The infectious dose can also be expressed as quanta concentration per unit volume of air (quanta/m^3^), taking into account cumulative exposure time and the dilution volume (e.g., the volume of a room). Infection risk estimation models use two approaches to estimate the quanta emission rate.

In the first approach, it can be modeled from a retrospective analysis of outbreaks [24,25,26,27,28].

In the second approach, the quanta emission rate is modeled from the predictive estimate of the infectious particle load expelled by patient zero [23,28,29,30].

Despite numerous scientific articles on the subject, it remains difficult to draw definitive conclusions and be quantitatively accurate about the transmissibility of epidemics. Indeed, the numerical values vary according to the contexts and studies. In particular, the lack of consensus on the complexity of a pathogen’s transmission and the multiplicity of influencing parameters leaves many questions open in this field [15].

#### 2.2.3. Environmental Conditions

Environmental conditions can also influence the spread of pathogens [31]. Enveloped viruses, such as COVID-19, are more sensitive to heat and ultraviolet light, making them less stable at high temperatures. Other viruses, such as influenza, are transmitted more efficiently at 5 °C than at 20 °C [32]. At 20 °C, the transmission efficiency of an influenza isolate exhibits a bimodal dependence on relative humidity (RH), with airborne (i.e., droplet or aerosol) transmission being maximal at 20–35% RH, poor at 50% RH, moderate at 65% RH, and absent at 80% RH [32]. Viruses also require a certain moisture level to survive and may become dehydrated and lose their ability to infect hosts in dry environments.

Bacteria can grow rapidly in hot, humid environments, such as in unrefrigerated food, but can also survive in cold, dry environments, such as stainless-steel surfaces. Temperature and humidity can also affect the ability of bacteria to produce toxins and infect hosts.

Fungi also have specific temperature and humidity requirements for their survival and growth. They can thrive in hot, humid environments, such as poorly ventilated bathrooms, but can also survive in dry, cool environments, such as cellars. Fungi can also produce spores that can be carried through the air and inhaled, causing allergies and fungal disease.

In summary, viruses, bacteria, and fungi have specific temperature and humidity requirements for their survival and ability to infect hosts. Hot, humid environments can promote the growth of bacteria and fungi, while cold, dry environments can enhance virus survival.

#### 2.2.4. Survival Time Outside the Host

The survival time of pathogens outside the host refers to the duration during which a pathogen can survive in the environment, i.e., on surfaces, in the air, or in water, without the presence of a host. Survival time varies with the type of pathogen, temperature, humidity, and type of surface [32,33,34]. Methods used to measure pathogen survival may include cell culture, staining assays, immunofluorescence assays, and quantitative PCR. The survival time of pathogens is usually expressed in hours or in days. For instance, the SARS-CoV-2 virus, responsible for the COVID-19 pandemic, can survive on stainless-steel surfaces for up to 72 h, on cardboard for up to 24 h, and on copper for up to 3 h.

#### 2.2.5. Emission

The last relevant characteristic is emission, which is defined as the speed at which an individual loads the air with pathogens. The emission factor is measured in quanta per hour (quanta·h^−1^). A quantum corresponds to the dose of aerosol sufficient to infect 63% of susceptible individuals. Emission factors depend on the symptomatic nature of the infection, physical activity, and vocalization. For SARS-CoV-2, the emission rate was assessed as varying between 1 quanta·h^−1^ and 100 quanta·h^−1^ by measuring the viral concentration in saliva [23], or between 0.37 quanta·h^−1^ and 32 quanta·h^−1^ using Monte-Carlo simulations [29]. In general, symptomatic people have a higher ability to spread pathogens through the air, mainly through coughing and sneezing. Studies have shown that coughing and sneezing can spread droplets at a speed of up to 50 m per second, while speech can spread smaller droplets at a speed of up to 10 m per second. However, it is important to note that airborne pathogen transmission also depends on the precautions taken to prevent spread, such as wearing masks, adequate ventilation of enclosed spaces, physical distancing, and hand hygiene [35,36].

#### 2.2.6. Severity of Disease

The severity of an emerging infectious disease is generally measured by three criteria: the proportion of hospitalized cases, the proportion of hospitalized cases in intensive care, and deaths. Data on hospital admissions and intensive care are not always available in real time. The calculation of mortality rates is subject to reporting biases, both for the number of deaths and for the number of confirmed or unconfirmed cases. For example, when only symptomatic cases, or even hospitalized cases, are reported, a much higher rate is obtained than if the entire population is screened, including paucisymptomatic (only mild symptoms) or even asymptomatic cases. More reliable estimates can be obtained from more localized epidemics, which allow for an almost systematic screening of cases and the exhaustive recording of deaths. This was the case on cruise ships (the Diamond Princess being the first) or military ships (such as the Charles de Gaulle aircraft carrier) during the COVID-19 pandemic [21].

The study of excess mortality helps to overcome these potential biases. Excess mortality is a statistical concept based on comparing observed mortality rates during an epidemic with mortality rates recorded in previous years [21].

## 3. Methodology

A list of diseases was compiled to better understand each disease (symptoms, incubation period, contagiousness, etc.) and to provide details about the mode of contamination. The selection of specific pathogens for analysis was guided by several key criteria to ensure their relevance to naval environments. Priority was given to pathogens known for their ability to spread in confined settings such as ships (e.g., COVID-19, influenza, and tuberculosis). The selection criteria also considered pathogens that were historically involved in outbreaks aboard military and civilian vessels, in order to reflect real-world risks. Infectious agents demonstrating resilience against standard prevention measures, such as norovirus, were also included due to their high transmissibility and containment challenges. To determine the probability of occurrence, a multi-faceted approach was employed, integrating epidemiological data, studies from analogous environments (e.g., hospitals and military bases), and expert assessments from specialized medical professionals. Diseases that would not be encountered in naval military environments due to their mode of transmission were excluded from the study (e.g., Bilharzia). In addition, the French Military Health Service (SSA) vaccination schedule was also considered [37]. Naval military ships are differentiated from civilian ships by their construction and purpose. Military ships are designed to be highly efficient, reliable, and scalable, incorporating the latest technologies in restricted spaces to enable marines to protect their country’s sovereignty across the globe. The military ships currently used by France include aircraft carriers, surface combatant vessels, patrol vessels, amphibious helicopter carriers, and submarines. To fulfill their missions, these military ships may need to make stopovers in various parts of the world, making them vulnerable to epidemics, hence the importance of ensuring their resilience to such threats. The diseases included in the analysis were classified according to a risk prioritization index that considers both clinical severity and contagiousness.

Both clinical severity and contagiousness were classified into five categories (Table 1 and Table 2), based on a literature review. The relevance and application of the rating grids were then validated by professionals in the medical field.

In our study, despite non-constant but fluctuating values, and in order not to complicate the analysis, we have chosen to use R_0_ as a measure of contagiousness.

By combining the contagiousness level and severity of a disease, we can classify epidemics by their level of risk. This approach is similar to the one used by the Institute for Disease Modeling at Harvard University. The risk prioritization index (RPI) was also calculated by multiplying clinical severity by contagiousness (Table 3). Since French sailors were vaccinated against a number of diseases, according to the SSA vaccination schedule [37], a coefficient of less than 1 can be applied to the RPI to account for vaccination. This coefficient is based on the assumption that sailors are vaccinated before the deployment, ensuring maximum vaccination coverage:A coefficient of 0.2 was applied to diseases with a vaccine effectiveness rate between 66% and 100% [38,39,40,41,42,43,44,45,46];A coefficient of 0.5 was applied to diseases with a vaccine effectiveness rate between 35% and 65% [47]; andA coefficient of 0.8 was applied to diseases with a vaccine effectiveness rate between 0 and 34%.

Risks are classified into three categories, based on the importance of the military ship’s mission:Category 1 (10 ≤ RPI ≤ 25): used for diseases with a significant impact on the military ship’s mission;Category 2 (5 ≤ RPI ≤ 9): used for diseases with a moderate impact on the military ship’s mission; andCategory 3 (1 ≤ RPI ≤ 4): used for diseases with a low impact on the military ship’s mission.

Several agents can be used to intentionally infect people, leading to a significant number of illnesses and deaths. This act is known as bioterrorism. Bioterrorism agents can be classified into three categories:Category A: high-priority agents, which include organisms that pose a significant risk to national security due to several factors:They are easily disseminated or transmitted from person to person;They have high mortality rates, with the potential for a major public health impact;They have the ability to cause public panic and social disruption; andThey require specific enhancements in disease surveillance.Category B: second-highest priority agents, which include pathogens that:Are moderately easy to disseminate;Cause moderate morbidity rates and low mortality rates; andRequire specific enhancements in disease surveillance.Category C: third-highest priority agents, which include emerging pathogens (such as Nipah virus and Hantavirus) that could potentially be engineered for mass dissemination in the future, due to:Their availability;Their ease of production and dissemination; andTheir potential for high morbidity and mortality rates, and their significant health impact.

Some infections considered to be bioterrorism threats are classified and have limited data available in the literature. As a result, only a few examples of such infections have been presented in Table 4.

## 4. Results

A total of fifty-eight infectious diseases were listed and described. Table 4a presents diseases related to viruses, Table 4b shows those related to bacteria, Table 4c shows those related to parasites, and finally, Table 4d shows those related to fungi.

Only diseases with a probable occurrence onboard were assessed in terms of clinical severity and contagiousness. The evaluation of contagiousness and clinical severity, which was reviewed by three medical doctors (A.G., F.L., and P.P.), including an infectious disease specialist (A.G.), remains a subjective assessment based on the scientific literature.

The highest-risk diseases in a military naval environment share several characteristics in terms of contagiousness and severity. They can be categorized into three main modes of transmission: direct contact with bodily fluids (Ebola, rabies, smallpox, and hepatitis E), fecal–oral transmission through contaminated water or food (cholera, Shiga toxin-producing bacteria, shigellosis, and hepatitis E), and airborne or droplet transmission (meningitis, pneumonic plague, tuberculosis, and smallpox). Among them, smallpox, pneumonic plague, and tuberculosis are particularly dangerous due to their airborne transmission in confined spaces, facilitating rapid contagion aboard a ship. Similarly, fecal–oral diseases such as cholera and shigellosis pose a major risk in an environment where potable water and food supplies are centralized, increasing the potential for outbreaks. Some diseases, such as Ebola and rabies, are less contagious but extremely lethal. Finally, tuberculosis and hepatitis E can develop into chronic infections, posing a persistent risk to crews. In a military naval setting, where close quarters and limited resources make it difficult to isolate infected individuals and implement rapid countermeasures, these diseases represent a significant threat to operational health and readiness.

## 5. Discussion

This literature review describes various diseases that can spread aboard military ships, and it emphasizes the characteristics of those pathogens that contribute to their transmission. Key among these characteristics are the reproduction number (R_0_) and the contagiousness, which together enable the definition of a risk prioritization index (RPI). The ultimate goal of this study is to minimize the operational impact on military boat missions by identifying those pathogens that pose the greatest risk. Based on the analysis conducted, pathogens can be classified according to four main groups: emerging diseases (e.g., COVID-19), serious diseases with low contagiousness (e.g., hepatitis), highly contagious diseases with low severity (e.g., varicella), and bioterrorism threats (e.g., smallpox).

Other similar studies on infectious diseases in confined or enclosed environments, particularly on military ships, cruise ships, and certain facilities like hospitals and detention centers, have already been conducted. These environments share common characteristics, such as restricted spaces, limited ventilation conditions, and a high number of close interactions, which facilitate the spread of pathogens. Previous research has examined infectious disease outbreaks on ships, particularly of respiratory viruses like SARS-CoV-2 and influenza, showing that airborne transmission is a major mode of spread in these contexts [48]. For example, the study by Rocklöv et al. (2020) [49] on COVID-19 aboard the Diamond Princess cruise ship demonstrates that, despite high ventilation rates and the absence of air recirculation, aerosol transmission remained dominant. This study highlights the difficulty of containing these pathogens in confined environments and underscores the limitations of traditional ventilation measures in these spaces. Our review corroborates these findings by identifying the key characteristics of pathogens that promote contagion in naval environments, such as the stability of airborne particles and the basic reproduction number (R0). Similarly, a study by Whittaker et al. (2004) [50] on viral gastroenteritis outbreaks aboard military ships showed that strict control measures, like enhanced ventilation and surface disinfection, are crucial, but are sometimes insufficient against interpersonal transmission in restricted spaces.

A crucial aspect requiring further exploration is how specific naval environmental conditions influence pathogen transmission dynamics. Unlike other enclosed settings such as hospitals or office buildings, military ships present unique challenges. For example, while ships are equipped with advanced ventilation systems, the efficiency of these systems in filtering airborne pathogens varies. Several studies [51,52] have indicated that even with high ventilation rates, aerosolized pathogens can persist and spread across compartments, necessitating additional control measures such as localized air filtration and UV-based disinfection systems. In addition, the high-density living conditions on military ships, coupled with frequent social and operational interactions, create ideal conditions for rapid disease transmission. Close-quarters sleeping arrangements, shared dining spaces, and communal hygiene facilities all contribute to the increased risk of outbreaks. Furthermore, the size of naval vessels varies depending on the type of ship, and the number of crew members on board also depends on the specific vessel. As a result, the available space per sailor in square meters will differ according to ship type. For instance, the level of proximity among crew members is much higher on a submarine than on an aircraft carrier. Submarines are designed to operate in confined spaces, meaning that sailors have very limited personal space. In contrast, aircraft carriers, being significantly larger, offer a larger volume per person, thereby reducing the risk of contamination. In addition, military ships often operate in isolated environments for extended periods, unlike land-based facilities where infected individuals can be quickly isolated or evacuated. This limited access to external medical resources requires a proactive approach, including onboard diagnostic capabilities and contingency planning for medical evacuations.

Some pathogens also present a greater challenge in naval environments due to their ability to spread through multiple transmission routes, making mitigation strategies more complex. For example, COVID-19 spreads via airborne aerosols, droplets, contaminated surfaces, and, potentially, fecal–oral transmission, requiring multiple layers of prevention. Norovirus, a common cause of viral gastroenteritis, is transmitted through direct contact, contaminated food/water, and airborne particles from vomiting, making control measures such as strict hygiene and disinfection essential. Influenza primarily spreads through aerosols but can also be transmitted via contaminated surfaces; therefore, it requires combined strategies of vaccination, ventilation, and surface cleaning. These examples highlight the complexity of disease control aboard military ships, where the confined nature of the environment amplifies the difficulty of limiting transmission.

In military contexts, several studies have focused on managing biological threats and emerging pathogens. For instance, the risk prioritization model in this study, which combines clinical severity and the reproduction potential index (RPI), is similar to methodologies used by the Harvard University Institute for Disease Modeling [53] to rank biological agents based on their spread potential in military settings. These approaches underscore the importance of prioritizing pathogens with high reproduction potential and those representing significant threats to military missions due to their rapid spread.

Airborne transmission has also been observed during outbreaks of influenza and other respiratory viruses on planes, which share similar confined, high-density environments with ships. These findings emphasize the challenges in containing pathogens in such environments, reinforcing the importance of understanding how pathogens propagate through the air in confined spaces [48]. Research conducted in hospitals also indicates that infection spread is often controlled through monitoring and rapid response, as well as measures such as isolating infected patients [54]. However, naval environments, due to their mobility and lack of equivalent medical facilities, require additional approaches, including advanced detection technologies and the integration of containment measures during ship design.

Recent research has criticized the use of the R0 index as a standard measure of contagiousness, especially in closed environments where factors such as population density, close interactions, and ventilation can influence epidemiological outcomes [55]. This study could benefit from models that incorporate additional parameters, such as crew movement analysis and quantum emissions in shared spaces. Previous work by Hsu et al. (2023) [56] shows that models based on these factors produce more accurate predictions for outbreaks in confined environments, such as cruise ships and military bases.

This approach underscores the complexity of managing disease risks on military ships and highlights several limitations. One limitation is the use of R0, which is often criticized due to the lack of consensus within the scientific community regarding its estimation [19]. The subjectivity of the evaluation of clinical severity and contagiousness, although based on scientific publications, is also a limitation of this study. Furthermore, while the criteria discussed in this study provide a foundation for simulating pathogen spread, understanding crew activities will be crucial for accurate transmission modeling. This activity analysis must be integrated as a parameter of these simulations. Indeed, several studies have shown that the majority of infections are transmitted between passengers [57]. Therefore, an analysis of crew movements—who performs which tasks, as well as where and when—will need to be integrated into these simulations.

The next steps of this work include 1) reducing the list of biological risk agents based on their common characteristics, and 2) developing methods and tools to assess, simulate, and improve the resilience of military ships [58].

Better identification and understanding of pathogens will improve detection (e.g., PCR tests and air quality measurements), prevention (e.g., an awareness of barrier gestures and isolation), and elimination (e.g., air or water decontamination). These measures can be taken into account in the design of ship architecture, with technological monitoring and cost-benefit analysis guiding decisions. Currently, several advanced solutions are being studied for implementation on military vessels, with all the integration constraints that this entails. For example, high-efficiency particulate air (HEPA) filters and ultraviolet germicidal irradiation (UVGI) are being tested to reduce the airborne transmission of pathogens. These systems can be incorporated into ventilation networks to provide continuous air purification. Among other technologies, the development of onboard biosensors capable of detecting airborne or waterborne pathogens in real time is a key area of research. These sensors use molecular detection techniques such as CRISPR-based assays or rapid PCR diagnostics. Self-disinfecting surfaces, antimicrobial coatings, and automated UV-C robots can also be used to reduce contamination risks in high-contact areas. Finally, improvements in ballast water management and onboard water purification technologies, such as ozone- and plasma-based disinfection, are being evaluated to prevent the spread of waterborne pathogens. Regarding the crew, future ship designs may include dedicated isolation spaces equipped with negative pressure systems to contain infectious diseases more effectively. Health monitoring systems could also analyze crew health data, detect early signs of disease outbreaks, and optimize response strategies through predictive modeling.

Rooms identified through simulation models and activity analysis could benefit from dedicated approaches such as air filtration, systematic surface disinfection, non-ionizing electric fields [59], or the use of far-UVC lamps to significantly reduce the risk of pathogen spread.

This study provides a foundation for further assessing the spread of pathogens aboard military ships. The results can be refined to target those pathogens that have the greatest impact on operational missions and that require appropriate countermeasures. Additional studies are needed to identify common characteristics among the pathogens of interest. These characteristics will then help to determine effective control strategies, which can be evaluated through:Experimental trials in a controlled environment to test the effectiveness of detection, identification, containment, and elimination solutions adapted to the constraints of the maritime environment (humidity, salinity, and vibrations); andNumerical simulations (particularly computational fluid dynamics) to model the spread of pathogens and assess preventive and detection measures.

This study highlights the high risk of airborne transmission and emphasizes the importance of studying this mode of transmission in future research. In addition, an analysis of sailors’ activities on board would help identify practices and times of day that promote disease transmission. However, since access to ships is restricted due to military confidentiality, the use of agent-based simulation models would be helpful.

## 6. Conclusions

In light of the results from similar studies, this review emphasizes that managing infectious diseases in military naval settings requires specific solutions, such as modeling crew interactions and integrating containment measures into the design of the ships themselves. The adoption of measures inspired by studies of other closed environments could also strengthen the biological resilience of military ships, helping to better anticipate and contain outbreaks. By developing early detection strategies and tailoring interventions to the characteristics of military ships, this study could also inspire more robust epidemic prevention policies, thereby minimizing operational risks.

## Figures and Tables

**Figure 1 ijerph-22-00572-f001:**
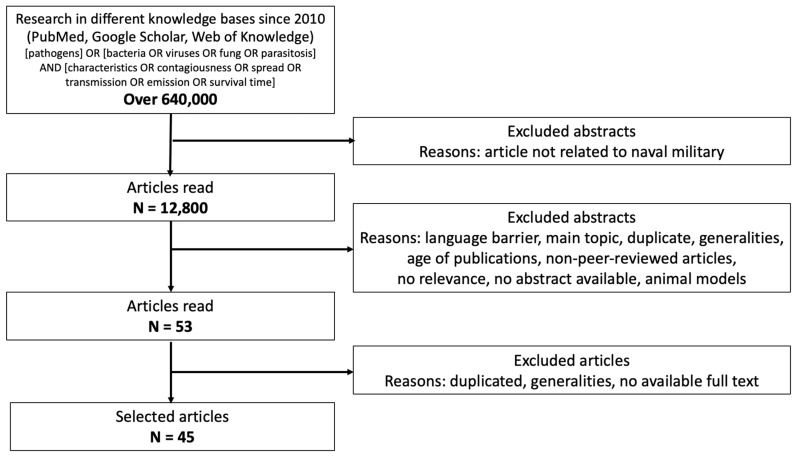
Flow chart illustrating how research into the identified articles was performed, following the PRISMA guidelines.

**Table 1 ijerph-22-00572-t001:** Classifications for clinical severity.

Clinical Severity	
5—Catastrophic	Frequent mortality
4—Critical	Immediate morbidity with severity
3—Major	Immediate morbidity without serious but with distant consequences
2—Moderate	Immediate morbidity without serious or long-term consequences
1—Minor	No clinical impact (ambulatory without impact on daily life)

**Table 2 ijerph-22-00572-t002:** Ratings for the contagiousness index (R0).

Contagiousness	
5—Extreme	R0 between 12.1 and 20
4—Very high	R0 between 7.1 and 12
3—Significant	R0 between 5.1 and 7
2—Moderate	R0 between 1.1 and 5
1—Minor	R0 between 0 and 1

**Table 3 ijerph-22-00572-t003:** Risk prioritization index. Clinical severity: 5—catastrophic; 4—critical; 3—major; 2—moderate; 1—minor. Contagiousness: 5—extreme; 4—very high; 3—significant; 2—moderate; and 1—minor.

		Contagiousness				
	Criteria	5	4	3	2	1
Clinical severity	5	25	20	15	10	5
	4	20	16	12	8	4
	3	15	12	9	6	3
	2	10	8	6	4	2
	1	5	4	3	2	1

**Table 4 ijerph-22-00572-t004:** (**a**): Diseases related to viruses, with their descriptions and risk prioritization calculations. Diseases that are not encountered in the military naval environment are listed, but they have not been classified according to the risk prioritization index (the boxes have been grayed out). (**b**): Diseases related to bacteria, with their descriptions and risk prioritization calculations. Diseases that are not encountered in the military naval environment are listed, but they have not been classified according to the risk prioritization index (the boxes have been grayed out). (**c**): Diseases related to parasites, with their descriptions and risk prioritization calculations. Diseases that are not encountered in the military naval environment are listed, but they have not been classified according to the risk prioritization index (the boxes have been grayed out). (**d**): Diseases related to fungi, with their descriptions and risk prioritization calculations. Diseases that are not encountered in the military naval environment are listed, but they have not been classified according to the risk prioritization index (the boxes have been grayed out).

(**a**)								
**Disease**	**Transmission**	**Probability of Occurrence in the Naval Military**	**Severity**		**Contagiousness**		**Moderating Coefficient**	**Risk Prioritization Index**
			**Symptoms**	**Rated**	**Description**	**Rated**		
Chikungunya	Transmission by mosquito	The disease is not very widespread in metropolitan France, but is regularly encountered in epidemic episodes throughout the world. Contamination is by mosquito bite, as well as human-to-human transmission by blood transfusion; its moderate contagiousness limits the probability that the disease can be found on board a military ship.	- High fever - Headache - Joint and muscle pain	3	It is not a contagious disease. Transmission of the virus occurs only through the mosquito bite.	1		3
COVID-19	Airborne transmission Hand-carried transmission	Very large number of cases and deaths worldwide. The varied and incapacitating symptoms, coupled with very simple modes of transmission, moderate contagiousness, short incubation period, and the history of numerous previous outbreaks in military buildings, suggest that new outbreaks are highly likely in the future.	- Fever - Headache - Chills - Coughing - Aches and pains - Fatigue - Sudden loss of smell (without nasal obstruction) - Complete loss of taste - Diarrhea - Difficulty breathing - Death	4	- By aerosol, from the mouth or nose. - From the hands when touching a surface soiled by droplets.	2	0.2	3.2
Dengue	Transmission by mosquito Transmission from mother to child	Despite a very large number of cases worldwide each year and significant symptoms with a high mortality rate, human-to-human transmission can only occur through specific means (mosquito bite or the transplacental route). This strongly limits the probability of the emergence of this disease on board.	- Severe abdominal pain - Vomiting - Bleeding gums or nose - Fatigue - Strong feeling of thirst - Pale, cold skin - Feeling of weakness	3	It is not a contagious disease. Transmission of the virus occurs only through the mosquito bite.	1		3
Ebola	Animal transmission Hand-carried transmission	This virus, which has extremely severe symptoms and a very high mortality rate, occurs in epidemic episodes. The moderate contagiousness and hand-carried transmission, combined with an incubation period of several days, may suggest that an epidemic could occur, but this would mainly follow a stopover in a hard-hit area.	- Fever - Headaches - Fatigue - Muscle pain and weakness - Vomiting - Diarrhea - Redness on the body - Bleeding (nose, gums, and bruises) - Blood in the urine or stool - Hemorrhages - Death	5	A person can contract the Ebola virus if he or she comes into contact with the bodily fluids of an infected person, or a surface or object contaminated with the bodily fluids of an infected person.	2		10 (TBR A)
Hepatitis A virus	Fecal–oral transmission	Despite moderate contagiousness, the disease is present throughout the world and can be spread between humans in a simple manner. Because symptoms take time to appear, the time it takes to detect the disease can allow for significant contamination in a ship’s crew. The disease is preventable with vaccination.	- Fever - Fatigue - Loss of appetite - Digestive disorders - Nausea - Joint pain - Hives	5	- Directly (contaminated hands, contact), from individual to individual. - Indirectly, through: - Water contaminated by viruses present in the stool of people with hepatitis A. - Contaminated food.	2	0.2	2
Hepatitis B virus	Sexual relations Blood transmission	Although the hepatitis B virus (HBV) is present in most of the biological fluids of infected people and is highly contagious, its probability of occurrence in a military naval environment is zero. (All sailors are vaccinated).						
Hepatitis C virus	Blood transmission	Hepatitis C is an infectious liver disease that is transmitted primarily through the bloodstream. It is caused by the HCV virus. Acute hepatitis C becomes chronic in approximately 70% of cases. With treatment, chronic hepatitis C is cured in 99% of cases. In the absence of treatment, liver fibrosis eventually appears. The incubation period for hepatitis C is usually 2 to 12 weeks. Acute hepatitis C is usually asymptomatic and about 30% of infected people clear the virus within 6 months of infection and recover without any treatment. Approximately 70% of infected people do not clear the virus and develop chronic hepatitis C. Chronic hepatitis C is responsible, in the shorter term, for cancer of the liver and cirrhosis in 15 to 30% of cases after several years of development.						
Hepatitis D virus	Blood transmission Skin lesion	Hepatitis D is an inflammation of the liver caused by HBD, which requires HBV to replicate. Hepatitis D cannot occur in the absence of hepatitis B.						
Hepatitis E virus	Fecal–oral transmission (contaminated water)	This virus is excreted in the stool of infected people and enters the human body through the intestines. It is transmitted mainly through contaminated drinking water. HEV infection is found all over the world. The disease is common in low- and middle-income countries with limited access to clean water, sanitation, hygiene, and health services.	- Moderate fever - Decreased appetite - Nausea and vomiting - Abdominal pain - Itching, rash, and joint pain - Jaundice - Dark urine - Light stools	5	- Water contaminated by viruses - Contaminated food	2		10
Human immunodeficiency virus	- Blood transmission - Sexual intercourse - Transmission from mother to child	A disease that is widespread throughout the world. The fact that transmission can occur through several modes and that symptoms take several weeks to appear increases the risk of multiple cases emerging within the naval forces.						
Influenza A H1N1	Animal transmission Airborne transmission Hand-carried transmission	This disease was widespread during the 2009 epidemic. The disease has a short incubation period and moderate contagiousness and it is preventable with the seasonal influenza vaccine, the effectiveness of which may vary, depending on the years.	- Fever - Headache - Coughing - Sore throat - Aches and pains - Fatigue - Runny nose - Nausea - Vomiting - Diarrhea - Pneumonia - Respiratory failure	4	Influenza A H1N1 is caught like other forms of influenza. People who are sick or who are developing the disease will cough and sneeze, which spreads droplets containing the virus around them. Healthy people become infected either by breathing in these microscopic particles (if they are in close proximity to sick people in an enclosed environment) or by touching a contaminated surface and then touching their eyes, mouth, or inside their nose (which seems to be a common mode of transmission). On a contaminated surface, the influenza virus remains active for a few hours at most (usually about two hours).	2		8
Junin	Animal transmission Direct contact Inhalation Ingestion	The very low number of cases each year are associated with a very particular mode of human-to-human transmission and because of its minor contagiousness, the risk can be considered as negligible with respect to the military naval domain.	- Fever - Malaise - Muscle aches - Gastrointestinal symptoms - Severe hemorrhagic manifestations - Shock	4	The R0 of the Junin virus is high, requiring strict measures of active surveillance, contact tracing, quarantine, and social distancing to stop transmission	4		16 (TBR A)
Lassa fever	Animal transmission Hand-carried transmission	This disease is not widespread worldwide and is of low contagiousness (R0 < 1), which leaves little room for the accidental contamination of a seafarer. Nevertheless, its symptoms are critical and the incubation period is long, which may allow the disease to incubate on several hosts before developing on board ship, or perhaps following an act of bioterrorism.	- Fever - Nausea - Vomiting - Abdominal pain - Headache - Muscle pain - Joint pain - Generalized weakness - Edema - Hemorrhages - Pericardial and pleural effusions - Renal and hepatic failure - Death	5	The virus can be transmitted from human to human, mainly in a hospital context, by skin–mucosal contact with a patient’s biological fluids.	1		5 (TBR A)
Marburg virus disease	Animal transmission Hand-carried transmission	The number of annual cases of this disease is extremely low. Although the symptoms are severe and human-to-human transmission is variable, the contagiousness remains moderate. Isolated cases could occur on board in a crew, but an epidemic seems unlikely.	- High fever - Severe headache - Severe malaise - Muscle and joint pain - Fatigue - Gastrointestinal symptoms - Loss of appetite - Unexplained bleeding - Rash - Organ failure	5	Marburg virus disease is highly infectious but not very contagious. It spreads through direct contact with infected animals, particularly fruit bats, or through contact with the bodily fluids of infected individuals. The virus is not airborne and is considered not to be contagious before symptoms appear	1		5 (TBR A)
Measles	Airborne transmission	Measles is a disease that is still very present around the world and its clinical severity is catastrophic, with a high mortality rate. Moreover, it is transmitted between human beings by air, with a very high level of contagiousness. Its incubation period of a few days can, therefore, allow a large number of sailors to be contaminated before the first symptoms appear. The disease is preventable with a vaccine.	- Fever - Rhinitis - Conjunctivitis - Coughing - Fatigue - Skin rash - Complications (30% of cases) (acute otitis, laryngitis, and diarrhea) - Deaths	5	Measles is a highly contagious disease caused by a virus that is easily transmitted through coughing, sneezing, and nasal secretions. One person infected with measles can infect 15 to 20 people.	4	0.2	4
Meningitis (Neisseria meningitidis, Haemophilus influenzae, Streptococcus pneumoniae, Listeria monocytogenes)	Fecal–oral transmission	In this type of meningitis, symptoms known as meningeal syndrome (headache, photophobia, and vomiting) predominate, while the patient’s general condition remains unaltered. The disease is generally benign: in patients without immune deficiency, recovery is usually spontaneous. Recovery takes place within a few days, with no after-effects. It is usually caused by widespread viruses belonging to the enterovirus family.	- Intense headaches - Nausea or vomiting - Stiff neck - Severe aches and pains - Extreme fatigue - Neurological symptoms	3	Viral meningitis is the most common form, caused by a virus (enterovirus, which can spread throughout the body).	2	0.2	1.2
Mpox	Airborne transmission Hand-carried transmission	A global mpox outbreak began in May 2022, with cases increasing in the Democratic Republic of the Congo and spreading to previously unaffected areas. The outbreak is caused by an orthopox virus, primarily spreading through close contact, and has led to over 100,000 cases in 122 countries. During a 2022 outbreak, 146 military health system beneficiaries, including 118 active-duty personnel, were affected by confirmed or suspected mpox cases.						
Mumps	Airborne transmission Hand-carried transmission	Although the disease causes few cases each year in France and the symptoms are moderate, it is easily transmitted by air and by hand, with very high contagiousness. Its long incubation period can lead to an epidemic.	- Fever - Headaches - Significant fatigue - Loss of appetite - Parotitis - Salivary gland damage - Pain in the ears - Difficulty swallowing	2	A person can get mumps in the same way they can catch a cold: by coming into contact with saliva particles from someone who is coughing or sneezing, by sharing a drink with an infected person, by kissing an infected person, or by touching a surface that has been contaminated with the virus. Therefore, frequent hand-washing is important.	1	0.2	0.4
Rabies	Animal transmission	In spite of the important symptoms and a very long incubation period (which could leave time to contaminate many people), this disease can only be caught by animal contact or the transplant of contaminated organs. It is, therefore, logical to evaluate its severity in the maritime world as minor.	- Anxiety - Agitation - Coma - Death	5	The disease is most prevalent in Africa Asia, and Latin America, where dogs are the main vector of transmission to humans. In Europe, bats can be infected by the lyssavirus, which is different from that in dogs. The disease can be contracted during stopovers.	2		10
Rubella	Airborne transmission Manipulative transmission Transplacental transmission	The number of annual cases in France is extremely low. Nevertheless, its simple transmission, by air and by hand, its important contagiousness, and its long incubation period could be the origin of an epidemic on board a military ship.	- Moderate fever - Headaches - Aches and pains - Pharyngitis - Conjunctivitis - Enlarged lymph nodes behind the ears and in the neck - Skin rash	2	Contamination occurs through droplets of saliva from the upper airways containing the virus: - During coughing, sneezing, blowing of the nose, and contact with hands soiled by saliva. - During close contact with infected persons. - From objects contaminated by secretions from the nose or throat (toys, handkerchiefs, etc.). In pregnant women, the rubella virus is transmitted to the fetus through the placenta.	3	0.2	1.2
Sexually transmitted infection	Sexual relations Transmission from mother to child	Widespread infection worldwide, but with minor contagiousness. The incubation period and its simple transmission through sexual intercourse suggest that several sailors could become infected before the first symptoms appear.						
Smallpox	Airborne transmission Hand-carried transmission	Smallpox is a disease considered to be eradicated to this day. Despite this, strains are kept in laboratories. The disease is very aggressive, relatively contagious, and is transmitted between humans in a simple way, by air or by hand. Its use as a bioterrorism weapon would be catastrophic for a crew.	- Fever - Headaches - Back pain - Abdominal pain - Feeling sick - Skin rash - Pustules - Lung, brain, and bone damage - Death	5	Contagiousness rate R0 = 5 This disease has been eradicated in theory but could be a terrorist biological risk.	2		10 (TBR A)
Varicella	Airborne transmission Hand-carried transmission	The number of annual cases in France is significant, with relatively troublesome symptoms, even proving fatal in rare cases, with very high contagiousness and a short incubation period. A case on board a military ship could quickly lead to others in sailors who would never have contracted the disease in the past or in whom shingles could be triggered.	- Moderate fever - Headache - Mild cough - Skin lesions	2	The varicella virus is transmitted by direct contact with the skin and mucous membrane vesicles, as well as by the respiratory route, by the inhalation of saliva droplets.	4	0.2	1.6
Yellow fever	Mosquito transmission	The number of annual cases worldwide is significant, as are the symptoms. However, since the mode of transmission is limited to mosquito bites, it is unlikely that a seafarer will become infected. If this does occur, symptoms appear quickly and there would be no transmission to another crew member.						
(**b**)								
**Disease**	**Transmission**	**Probability of Occurrence in the Naval Military**	**Severity**		**Contagiousness**		**Moderating Coefficient**	**Risk Prioritization Index**
			**Symptoms**	**Rated**	**Description**	**Rated**		
Anthrax	Animal transmission	Few cases occur per year around the world and there is no transmission from one human being to another. Most outbreaks occur as a result of malicious acts. The most feared scenario for a military vessel would, therefore, be the use of the pathogen as a weapon of bioterrorism.	- Fever - Nausea - Vomiting - Loss of appetite - Abdominal pain - Diarrhea - Vomiting blood - Death	5	Anthrax cannot be transmitted from person to person, but in rare cases, cutaneous anthrax can be spread from person to person through direct contact with an infected person or an object contaminated by an infected person.	1		5 (TBR A)
Botulism	Contaminated food Contaminated water	In spite of its very serious symptoms, there are few cases of the disease each year in France or in Europe, and human-to-human transmission is impossible. Therefore, only isolated cases could appear following the consumption of contaminated food or contaminated water during stopovers in more affected areas of the world. As the incubation period is very short, the disease’s diagnosis and treatment are made easier.	- Ocular damage (lack of accommodation, blurred vision) - Dryness of the mouth - Defects in swallowing or even speech - Paresis or paralysis of muscles - Paralysis of the respiratory muscles - Death	5	Botulism is not transmitted from person to person. Botulism develops when a person ingests the toxin (or, rarely, if the toxin is inhaled or injected) or when the bacteria grow in the intestines and release the toxin.	1		5 (TBR A)
Brucellosis	Animal transmission Contaminated food	The number of cases worldwide each year is low, with common symptoms despite some rare deaths. As human-to-human transmission is extremely rare, only isolated cases could appear, with a very low probability.	- Fever - Sweating - Fatigue - Chills - Joint and muscle pain - Headache - Loss of appetite - Weight loss - Abdominal pain - Swelling of lymph nodes	3	Brucellosis is not highly contagious among humans, as it primarily spreads from animals to humans rather than from person to person. However, brucellosis can be transmitted through the inhalation of aerosols.	1		3 (TBR B)
Cholera	Contaminated food Contaminated water	This disease is quite widespread around the world; its mode of contamination is simple, as is its mode of human-to-human transmission. Associated with high contagiousness and a relatively short period of onset of symptoms, the probability that a sailor could be a carrier becomes important.	- Diarrhea - Vomiting - Cardiovascular collapse - Death - Severe dehydration	4	It is a contagious disease that is transmitted by dirty hands and contaminated food or water. Less than 25% of infected people develop symptoms and 10 to 20% of them will develop severe disease.	4		16 (TBR B)
Diphtheria	Airborne transmission Hand-carried transmission	Despite the catastrophic clinical severity, the number of annual cases is too low for the disease to be of concern in our field. All sailors are vaccinated and as a result, this risk can be excluded.	- Formation of membranes in the throat - Asphyxiation - Production of toxin - Weakening of the heart - Paralysis - Failure of vital organs - Death	5	Contagiousness index R0: 2.5	2	0.2	2
Erysipelas	Hand-carried transmission	The number of cases in France each year is low, and although moderately contagious, the disease is only transmitted by direct contact with an injured person, which strongly limits the probability of an epidemic developing on board a French ship.	- Fever - Chills - Feeling sick - Skin inflammation	2	Erysipelas is contagious and is transmitted through skin-to-skin contact if the uninfected person also has an “entry point” (cut, wound, etc.).	1		2
Glanders	Contaminated food Contaminated water Animal transmission	This disease develops very serious and even fatal symptoms. Despite this, human-to-human transmission is unlikely. Isolated cases are, nevertheless, possible following stopovers in areas where the disease has not been eradicated. Another case to consider is the use of this agent as a bioterrorism weapon.	- Fever - Diffuse pain - Bloody and then purulent nasal secretions - Skin ulcerations - Disseminated abscesses - Death	5	The infecting dose of these bacteria is low, and aerosolized transmission is possible. This property has already enabled these agents to be used as a weapon of war in the last century. The use of these bacteria in acts of bioterrorism cannot be ruled out. European recommendations exist for the treatment and prophylaxis of glanders. Human-to-human transmission is unlikely.	1		5 (TBR B)
Legionellosis	Respiratory (inhalation of contaminated water aerosol)	Despite the troublesome symptoms, the number of annual cases is limited and the disease is not contagious. Nevertheless, contamination of drinking water on board a ship could result in the contamination of many of the crew members.	- Fever - Chills - Coughing - Difficulty breathing - Nausea - Confusion	2	It is not a contagious disease from one person to another. Contamination is by the inhalation of contaminated water diffused in aerosol form.	1		2
Leptospirosis	Waterborne transmission (dermal contact of injured skin or mucosa only, with water infested with bacteria) Animal transmission (infected animal or rat bite)	Given the number of cases per year at the national and global levels, it is unlikely that a French sailor would be exposed to this disease. Although its clinical severity is evaluated as critical, its contagiousness is null because it is not transmissible from one human to another.	- High fever with chills - Headache - Muscle and joint pain It can progress to the kidney or liver and cause meningeal or pulmonary damage.	4	The bacteria mainly enter through damaged skin or mucous membranes.	1		4
Melioidosis	Skin contact (on burns or abrasions) Contaminated food Contaminated water Inhalation of contaminated aerosols	The clinical severity of this disease is catastrophic. Despite this, the number of annual cases worldwide is relatively low and the contagiousness is minor because human-to-human transmission is impossible. Rare individual cases may appear after visits to an affected area, but could not cause an epidemic.	- Pneumonia - Acute respiratory distress - Septicemia - Skin infection (following inoculation or contact with a wound): pustules, abscesses, and ulcerations - Deep abscesses, which can affect all organs including the brain - Septic arthritis and osteomyelitis - Urinary infection and prostatitis - Otitis - Death	5	Humans can contract melioidosis by contamination from scrapes or burns and by ingestion or inhalation, but not directly from infected animals or people.	1		5 (TBR B)
Meningitis (Neisseria meningitidis, Haemophilus influenzae, Streptococcus pneumoniae, Listeria monocytogenes)	Airborne transmission Contaminated food	Despite the troublesome symptoms and a rather simple mode of transmission, the number of annual cases is relatively limited worldwide and the contagiousness is moderate. One can easily conclude that the potential seriousness of such a disease on board a ship will also be moderate.	- Intense headaches - Nausea or vomiting - Stiff neck - Severe aches and pains - Extreme fatigue - Neurological symptoms	5	Meningococci are transmitted by close (at less than one meter), direct, and prolonged contact (more than one hour) with nasopharyngeal secretions. Promiscuity on board ship and lengthy missions can encourage disease transmission.	2	0.2	2
Pertussis	Airborne	Although the disease primarily affects children and the elderly, it can be contracted at any age and several times in a lifetime. Although the symptoms are mild, its airborne transmission and extreme contagiousness make it a potential concern in the naval context.	- Coughing fits, violent and repeated - Spasms - Difficulty breathing - Puffy, red, or blue face (lack of oxygen) - Bursting of small blood vessels around the eyes - Vomiting	2	Pertussis is highly contagious and it is estimated that one sick person can infect an average of 15 to 17 people. This contamination is via airborne contact with the sick person through droplets from the nose or mouth when coughing.	5	0.2	2
Plague	Airborne transmission Hand-carried transmission	Few cases occur worldwide each year. Despite this, the proportion of deaths is high, human-to-human transmission is simple, and the incubation period is short. The disease, therefore, is rapidly detectable. Nevertheless, a case on board a military ship could cause an epidemic.	- Fever - Buboes (lymph nodes draining the puncture site) - Septicemia - Bronchopneumonia - Bloody sputum - Death	5	Infection can be transmitted from human to human via respiratory droplets. Handling the bodies of people who have died of plague is another possible mode of contamination.	2		10 (TBR A)
Q Fever	Animal transmission	Given the number of cases per year at the national and global levels, it is unlikely that a French sailor would be exposed to this disease. Although its clinical severity is evaluated as being major, its contagiousness is null because it is not transmissible from one human to another.	- High fever - Severe headaches - Fatigue and weakness - Muscle and joint pain - Dry cough - Chest pain - Nausea, vomiting and diarrhea - Loss of appetite and weight loss	3	Q fever is highly contagious, but it does not spread easily from person to person. Instead, it is primarily transmitted from animals to humans through environmental exposure: - Inhalation of contaminated aerosols - Direct contact with infected animals or animal products - Consumption of contaminated dairy products - Blood transfusions or organ transplants - Sexual transmission	1		3 (TBR B)
Salmonellosis	Fecal–oral transmission	Food contamination on a French military ship cannot be ruled out. The rapid emergence of relatively incapacitating symptoms can even be fatal in rare cases, and its human-to-human transmission by a simple mode could lead to an epidemic on board.	- Fever - Diarrhea - Vomiting - Abdominal pain - Death	4	Contamination is strictly human-to-human for Salmonella. It occurs either directly, through dirty hands, or indirectly, through the ingestion of water or food soiled by the stool of a healthy carrier or a patient. A contaminated meal could infect many people.	1		4 (TBR B)
Scarlet fever	Airborne transmission Manipulative transmission	Scarlet fever has become a rare disease, affecting few individuals worldwide each year. Although moderately contagious and transmissible by air and by hand, the disease rarely affects adults, who are usually already immune.	- Fever - Chills - Angina - Difficulty swallowing - Swollen neck nodes - Headaches - Nausea - Vomiting - Abdominal pain	3	R0 contagiousness index: 4	2		6
Sexually transmitted infection	Sexual relations Transmission from mother to child	Widespread infection worldwide, but with minor contagiousness. The incubation period and its simple transmission through sexual intercourse suggest that several sailors could become infected before the first symptoms appear.						
Shiga toxin-producing E. coli (STEC)	Fecal–oral transmission	Few cases of this disease occur each year, but the clinical severity in case of infection is high. Although human-to-human transmission is rare, it occurs through a simple route, which is facilitated in communal spaces. It is therefore possible that a small outbreak of the disease could occur on board a military ship.	- Fever - Vomiting - Abdominal pain - Diarrhea (bloody) - Coagulation - High blood pressure - Hemolytic uremic syndrome - Death	5	Human-to-human transmission is possible, but rare. In the majority of cases, it is observed in the family environment or in communities (such as in nurseries).	2		10 (TBR B)
Shigellosis	Contaminated food Contaminated water Fecal–oral transmission	At the French Navy level, the probability of an epidemic of this disease is low. The number of annual cases worldwide is low and the contagiousness is moderate. Although the incubation period is short and the symptoms very severe, only isolated cases could eventually appear.	- Fever - Abdominal pain - Diarrhea - Blood in the stool - Hypoglycemia - Dehydration - Acute renal failure - Intestinal obstruction - Death	4	Shigella is transmitted by the fecal-–oral route (through the stool of infected patients or convalescent carriers): 10 to 100 bacilli are sufficient to cause the disease. Humans are the only reservoir and can eliminate these bacteria in their stool for weeks after a dysenteric episode.	3		12 (TBR B)
Tuberculosis	Airborne transmission	The disease is fairly widespread throughout the world and its airborne transmission makes it relatively easy to spread. Although moderately contagious, it can take several months to incubate, which may be enough time to infect a large number of sailors, eventually leading to the emergence of serious symptoms. This has already occurred in 2001 on aircraft carriers.	- Fever - Loss of weight - Prolonged cough - Death	5	Tuberculosis is transmitted by airborne microsecretions from a person with tuberculosis, especially when coughing, talking, singing, or sneezing. Only the respiratory forms (pulmonary, bronchial, and laryngeal forms) of tuberculosis disease are contagious. Extra-respiratory localizations are not contagious. Latent infection is not contagious.	2		10
Tularemia	Animal transmission	Very few cases of the disease occur each year, and there is no human-to-human transmission. Despite the troublesome symptoms and the long incubation period, even the probability of isolated cases is very limited.	- Fever and chills - Skin ulcer - Swollen and painful lymph nodes - Fatigue, headache, and muscle pain - Eye redness - Tearing and eye discharge - Sore throat and mouth ulcers - Difficulty swallowing - Abdominal pain, diarrhea and vomiting - Fatigue - Sepsis and multi-organ failure	4	Tularemia is highly infectious but is not easily contagious between humans. It primarily spreads from animals to humans through direct contact, insect bites, ingestion, or inhalation of contaminated materials.	1		4 (TBR A)
Typhoid fever	Fecal–oral transmission	Numerous cases are recorded each year around the world. As the disease can be incapacitating and even fatal (and, given its easy human-to-human transmission, simple, with a moderate R0 and a long incubation period), it is possible that an epidemic could occur following a stopover in a region where the disease is endemic.	- Fever - Headaches - Anorexia - Depression - Abdominal pain - Diarrhea or constipation - Bowel, heart, or brain complications - Death - Severe dehydration	5	Typhoid is spread by ingesting water or food contaminated with the stool of an infected person (fecal–oral transmission). The infected person remains contagious as long as he or she is excreting the bacteria in their stool. Two to five percent of infected people become chronic carriers.	3	0.5	7.5 (TBR B)
Typhus	Arthropod vectors (fleas, lice, and mites)	Typhus is transmitted simply by contact and its symptoms can be relatively severe. Nevertheless, the disease has been very rare in humans since the 2000s, and is, therefore, of little concern.	- Sudden high fever - Severe headache - Chills and shaking - Muscle and joint pain - Extreme fatigue and weakness - Confusion or delirium - Low blood pressure and organ failure - Nausea and vomiting	4	Typhus is not directly contagious between humans in most cases. Instead, it is transmitted through arthropod vectors (lice, fleas, mites, or ticks)	1		4 (TBR B)
(**c**)								
**Disease**	**Transmission**	**Probability of Occurrence in Naval Military**	**Severity**		**Contagiousness**		**Moderating Coefficient**	**Risk Prioritization Index**
			**Symptoms**	**Rated**	**Description**	**Rated**		
Bedbugs	Hand-carried transmission	Increasingly frequent cases in the national territory, but contagiousness is moderate.	- Itching - Skin redness	1	Due to the speed of reproduction, the bedbug can quickly infect a boat. Unlike ticks or mosquitoes, bedbugs present no risk of transmitting infectious agents (viruses, bacteria, parasites, etc.).	1		1 (TBR B)
Bilharzia	Water transmission	Despite a large number of cases worldwide and critical clinical severity, the transmission of this disease can only occur through contaminated water. This greatly limits the likelihood of an outbreak on board a military vessel.						
Cryptosporidiosis	Fecal–oral transmission	The number of cases of the disease is very low and its symptoms are limited, although troublesome. R0 contagiousness is minor, despite a common mode of transmission. A few isolated cases could eventually appear, but not enough to create an epidemic.	- Diarrhea - Abdominal cramps or pain - Nausea - Vomiting - Fever - Loss of appetite - Weight loss - Dehydration - Fatigue	2	Cryptosporidiosis is primarily transmitted through the fecal–oral route. The infection is highly contagious, particularly in environments where hygiene is poor or sanitation is inadequate.	2		2 (TBR B)
Integumentary leishmaniasis	Blood transmission Transmission from mother to child	It is unlikely that a contamination event will occur on board a French military vessel. The disease is not widespread worldwide, its transmission is carried out by particular vectors, and the incubation period varies from several weeks to several years.						
Malaria	Transmission by mosquito Blood transmission Transmission from mother to child	The probability of a sailor being affected is limited and contamination of other sailors is even more limited, due to the very specific modes of human-to-human transmission.						
Plasmodium falciparum malaria	Transmission by mosquito Blood transmission Transmission from mother to child	The probability of a sailor being affected is limited and the contamination of other sailors is even more limited, due to the very specific modes of human-to-human transmission.						
Scabies	Manipulative transmission Sexual intercourse	The number of annual cases worldwide is very high, but slightly less at the national level. Despite this, the disease is very contagious, its transmission is via simple vectors, and the incubation period of several weeks suggests that a large number of sailors could become infected before the first symptoms emerge.	- Itching - Scratchy lesions	1		4		4
Sexually transmitted infection	Sexual relations Transmission from mother to child	Widespread infection worldwide, but with minor contagiousness. The incubation period and simple transmission through sexual intercourse suggest that several sailors could become infected before the first symptoms appear.						
(**d**)								
**Disease**	**Transmission**	**Probability of Occurrence in Naval Military**	**Severity**		**Contagiousness**		**Moderating Coefficient**	**Risk Prioritization Index**
			**Symptoms**	**Rated**	**Description**	**Rated**		
Meningitis (Cryptococcus neoformans, Histoplasma capsulatum)	Animal transmission	Fungal meningitis occurs mainly in immunocompromised individuals and is caused by yeasts (microscopic fungi) such as cryptococci.	- Vigilance disorders - Cranial nerve paralysis	5	This type of meningitis generally occurs in people with serious illnesses or weakened immune systems.	1		5
Ringworm	Fungi Animal transmission Direct contact Hand-carried transmission	Cases of ringworm are very rare and mostly affect children. Although it is easily transmitted by skin-to-skin contact, symptoms are mild. Its rarity makes it a disease of minor concern in our field.						

## Data Availability

The data presented in this article are available on PubMed, Web of Knowledge, and Google Scholar, and are referenced at the end of this article.

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
