# Peer review of "Methodology of Epidemic Risk Analysis in the Naval Military"

_ijerph, 2025, doi:10.3390/ijerph22040572_

Round 1

Reviewer 1 Report

Comments and Suggestions for Authors

·     Clarify the criteria used for selecting specific pathogens for analysis, particularly the methodology for determining the probabilities of their occurrence in naval settings.

·        Although the study adheres to PRISMA guidelines, providing additional details about the selection process and exclusion criteria for the studies would enhance reproducibility.

·        The R0 values and clinical severity ratings are subject to contextual variability. Provide confidence intervals or sensitivity analyses which can enhance the robustness of these measures.

·        Justify the moderating coefficients applied for vaccination effectiveness with specific references to immunological studies.

·        Integrate countermeasure strategies and evaluate their effectiveness in naval settings to strengthen the discussion on bioterrorism.

·        Add some detailed case studies of historical outbreaks on naval ships which will enhance the practical relevance of the findings.

·        A discussion section addressing how specific naval environmental factors (e.g., ventilation systems, crew density, and mission duration) impact pathogen transmission dynamics can be added.

·        Expand the discussion on diseases that involve multiple transmission routes (e.g., airborne and fomite) and how they complicate mitigation strategies in naval settings.

Author Response

Comment 1 : Clarify the criteria used for selecting specific pathogens for analysis, particularly the methodology for determining the probabilities of their occurrence in naval settings.

Answer 1 : The selection of specific pathogens for analysis was based on several key criteria to ensure relevance to naval environments:

  • Transmissibility in Naval Settings: Priority was given to pathogens known for their ability to spread in confined environments such as ships, including COVID-19, influenza, and tuberculosis.
  • History of Outbreaks in Confined Spaces: Pathogens previously involved in outbreaks aboard military and civilian vessels were included to reflect real-world risks.
  • Resistance to Countermeasures: Infectious agents demonstrating resilience against standard prevention measures, such as Norovirus, were considered due to their potential for rapid transmission and difficulty in containment.

To determine the probability of occurrence, a multi-faceted approach was employed, combining epidemiological data, studies from certain environments (e.g., hospitals, military bases), and expert assessments from specialized medical professionals. This methodological triangulation ensured a comprehensive and evidence-based evaluation of pathogen risks in naval settings.

Moreover, basic reproduction number (RO) is a commonly used indicator to estimate the transmissibility of an infectious disease in a fully susceptible and non-immune population. In many epidemiological studies, it is accepted to use a general R₀ based on values from scientific literature for each disease. However, we are aware that R₀ can vary depending on several factors: location (population density, climate, infrastructure), the studied population (age, prior immunity, social behaviors), and specific transmission conditions (for example, a military ship with confined space versus a city).

These criteria have been added in methodology.

Comment 2 : Although the study adheres to PRISMA guidelines, providing additional details about the selection process and exclusion criteria for the studies would enhance reproducibility.

Answer 2 : Although this study adheres to PRISMA guidelines, additional details regarding the selection process and exclusion criteria are provided to enhance reproducibility. The study selection was conducted based on several key criteria:

  • Temporal Restrictions: Studies were selected within a defined publication timeframe to ensure the relevance of the findings.
  • Linguistic Limitations: Only studies available in accessible languages, with a preference for English-language publications, were considered.
  • Relevance to Naval or Maritime Environments: Priority was given to studies focusing on pathogen transmission, epidemics, and biological resilience specifically in maritime contexts.
  • Exclusion of Duplicate or Non-Relevant Studies: Duplicate records and studies with limited applicability to the research scope were systematically removed.
  • Emphasis on Peer-Reviewed Literature: To ensure scientific rigor, priority was given to peer-reviewed articles and high-quality studies.

These criteria were established to refine the dataset and ensure that the selected studies provide relevant and robust insights into disease transmission and mitigation strategies aboard naval and maritime vessels.

Comment 3: the R0 values and clinical severity ratings are subject to contextual variability. Provide confidence intervals or sensitivity analyses which can enhance the robustness of these measures.

Answer 3: The R0 values ratings are illustrated in Table II. This figure provides a detailed representation of the variability in these measures within the studied context.

Comment 4: Justify the moderating coefficients applied for vaccination effectiveness with specific references to immunological studies.

Answer 4: Specific references about moderating coefficients applied for vaccination effectiveness have been added in methodology section:

  • [38] Van den Boogaard J, de Gier B, de Oliveira Bressane Lima P, Desai S, de Melker HE, Hahné SJM, Vedhuijzen IK. Immunogenicity, duration of protection, effectiveness and safety of rubella containing vaccines: a systematic literature review and meta analysis. Vaccine, 2001, 39(6):889-900. doi: 10.1016/j.vaccine.2020.12.079
  • [39] Chit A, Zivaripiran H, Shin T, Lee JKH, Tomovici A, Macina D, Johnson DR, Decker MD, Wu J. Acellular pertussis vaccines effectiveness over time: a systematic review, meta-analysis and modelling study. Plos One, 2018, 13(6). doi: 10.1371/journal.pone.0197970
  • [40] Griffin DE. Measles Vaccine. Viral Immunology, 2018, 31(2):86-95. doi:10.1089/vim.2017.0143.
  • [41] Langan RC, Goodbred AJ. Hepatitis A. American Family Physician, 2021, 104(4):368-374.
  • [42] Pillsbury A, Quinn H. An assessment of measles vaccine effectiveness, Australia, 2006-2012. Western Pac Surveill ResponseJ, 2015, 6(3):43-50. doi: 10.5365/WPSAR.2015.6.2.007.
  • [43] Deeks SL, Lim GH, Simpson MA, Gagné L, Gubbay J, Kristjanson E, Fung C, Crowcroft NS. An assessment of mumps vaccine effectiveness by dose during an outbreak in Canada. Canadian Medical Association Journal, 2011, 183(9):1014-1020. doi: 10.1503/cmaj.101371
  • [44] Truelove SA, Keegan LT, Moss WJ, Chaisson LH, Macher E, Azman AS, Lessler J. Clinical and epidemiological aspects of diphtheria: a systematic review and pooled analysis. Clinical Infectious Diseases, 2019, 71(1):89-97. doi: 10.1093/cid/ciz808-     
  • [45] Brisson M, Edmunds WJ, Gay NJ. Varicella vaccination: impact of vaccine efficacy on the epidemiology of VZV. Journal of Medical Virology, 2003(70):31-37. doi: 10.1002/jmv.10317
  • [46] Quiambao B, Peyrani P, Li P, Cutler MW, Van Der Wielen M, Perez JL, Webber C. Efficacy and safety of a booster dose of the meningococcal A, C, W, Y-tetanus toxoid conjugate vaccine administrered 10 years after primary vaccination and long-term persistence of tetanus toxoid conjugate or polysaccharide vaccine. Human Vaccines Immunotherapeutics,2020, 16(6):1272-1279. doi: 10.1080/21645515.2020.1744363-
  • [47] Milligan R, Paul L, Richardson M, Neuberger A. Vaccines for preventing typhoid fever (Review). Cochrane Database of Systematic Reviews, 2018, 5. doi: 10.1002/14651858.CD001261.pub4

Comment 5: Integrate countermeasure strategies and evaluate their effectiveness in naval settings to strengthen the discussion on bioterrorism.

Answer 5: The aim of this work is to better understand the spread of diseases aboard military ships in order to limit it using hygiene measures, confinement, or new technologies. This research could therefore contribute to a strategy in naval design. Furthermore, there are countermeasure strategies specific to each navy, such as vaccination schedules, medical equipment, or food supplies.

Measures can be taken into account into design of ship are developed in the discussion: “Currently, several advanced solutions are being studied for implementation on military vessels with all the integration constraints that this entails. For example, High-Efficiency Particulate Air (HEPA) filters and Ultraviolet Germicidal Irradiation (UVGI) are being tested to reduce airborne transmission of pathogens. These systems can be incorporated into ventilation networks to provide continuous air purification. Among other technologies the development of onboard biosensors capable of detecting airborne or waterborne pathogens in real time is a key area of research. These sensors use molecular detection techniques such as CRISPR-based assays or rapid PCR diagnostics. There also exists self-disinfecting surfaces, antimicrobial coatings, and automated UV-C robots considered to reduce contamination risks in high-contact areas. Finally, improvement ballast water management and onboard water purification technologies, such as ozone and plasma-based disinfection, are being evaluated to prevent the spread of waterborne pathogens. Regarding the crew, future ship designs may include dedicated isolation spaces equipped with negative pressure systems to contain infectious diseases more effectively and health monitoring systems could analyze crew health data, detect early signs of disease outbreaks, and optimize response strategies through predictive modeling”.

Comment 6: Add some detailed case studies of historical outbreaks on naval ships which will enhance the practical relevance of the findings.

Answer 6: Some detailed case studies of historical outbreaks on naval ship have been added:

“The study by De Laval et al. [9] showed that a large proportion of the crew of the French aircraft carrier Charles de Gaulle was infected, highlighting the rapid spread of SARS-CoV-2 in a confined environment. Crowded conditions, shared ventilation, and frequent interactions among sailors facilitated the transmission of the virus. The majority of cases were symptomatic, but most presented mild to moderate forms, with only a limited number of individuals requiring hospitalization. Nevertheless, despite attempts at isolation and social distancing on board, the virus spread before strict measures could be implemented, leading to the mission being cut short. On the USS Theodore Roosevelt aircraft carrier, more than 1,200 sailors were infected, representing approximately 25% of the crew. A portion of the crew was quarantined in Guam, which had a major impact on the ship's operational readiness."

Comment 7: A discussion section addressing how specific naval environmental factors (e.g., ventilation systems, crew density, and mission duration) impact pathogen transmission dynamics can be added.

Answer 7: The discussion has been completed. However, it is difficult to provide precise values on the density of people per square meter as this information can be confidential:

 A crucial aspect that requires further exploration is how specific naval environmental conditions influence pathogen transmission dynamics. Unlike other enclosed settings such as hospitals or office buildings, military ships present unique challenges. For example, while ships are equipped with advanced ventilation systems, the efficiency of these systems in filtering airborne pathogens varies. Several studies [50,51] have indicated that even with high ventilation rates, aerosolized pathogens can persist and spread across compartments, necessitating additional control measures such as localized air filtration and UV-based disinfection systems. Next, the high-density living conditions on military ships, coupled with frequent social and operational interactions, create ideal conditions for rapid disease transmission. Close-quarters sleeping arrangements, shared dining spaces, and communal hygiene facilities all contribute to the increased risk of outbreaks. Furthermore, the size of naval vessels varies depending on the type of ship, and the number of crew members on board also depends on the specific vessel. As a result, the available square meters per sailor differ according to the ship type. For instance, the level of proximity among crew members is much higher on a submarine compared to an aircraft carrier. Submarines are designed to operate in confined spaces, meaning that sailors have very limited personal space. In contrast, aircraft carriers, being significantly larger, offers a larger volume per person, thereby reducing the risk of contamination. To finish, unlike land-based facilities where infected individuals can be quickly isolated or evacuated, military ships often operate in isolated environments for extended periods. This limited access to external medical resources requires a proactive approach, including onboard diagnostic capabilities and contingency planning for medical evacuations.

Comment 8: Expand the discussion on diseases that involve multiple transmission routes (e.g., airborne and fomite) and how they complicate mitigation strategies in naval settings.

Answer 8: The discussion has been completed:

 Additionally, some pathogens present a greater challenge in naval environments due to their ability to spread through multiple transmission routes, making mitigation strategies more complex. For example: Covid-19 spreads via airborne aerosols, droplets, contaminated surfaces, and potentially fecal-oral transmission, requiring multiple layers of prevention; norovirus, a common cause of viral gastroenteritis, transmits through direct contact, contaminated food/water, and airborne particles from vomiting, making control measures like strict hygiene and disinfection essential; influenza which primarily spread through aerosols but can also be transmitted via contaminated surfaces, require combined strategies of vaccination, ventilation, and surface cleaning. These examples highlight the complexity of disease control aboard military ship, where the confined nature of the environment amplifies the difficulty of limiting transmission.

Reviewer 2 Report

Comments and Suggestions for Authors

Thank you for the opportunity to review this manuscript. This paper reviews epidemiologic risks of disease transmission onboard naval vessels. The paper includes an interesting discussion of modes of disease transmission. I would encourage inclusion in the discussion of more information regarding disease mitigation onboard ships, not just a list of diseases. The list of diseases is interesting but inconsistent in quality, mentions effective vaccines only on occasion, and makes some assumptions about risk that may not be accurate.

Under "Literature review", the authors indicated finding only 65 papers using their selected search terms. When I enter their search terms into PubMed alone, I find over 650,000 entries. I assume that there were more criteria used in their search than described.

Under 2.1.1, the authors write that "the greatest risk of pathogen transmission remains airborne transmission." In what way do they mean "the greatest risk"? Is this a reference to the severity of the diseases commonly transmitted through the airborne route, or is it a reference to the relative contagiousness? (Norovirus, while not a particularly dangerous disease in military populations, is widespread on ship, spreads rapidly, and carries a high risk for outbreaks.)

Under 2.1.2, the authors describe R0 as a key predictor of contagiousness. I would argue that R0 predicts nothing; rather, it is a mathematical approximation of observed disease transmission. I would consider rephrasing that sentence to reflect that R0 is a way to quantify contagiousness, not predict it per se. (The authors' subsequent description of how R0 is derived is very well-written, though.)

There is an error in the numbering of sections; section "2. Methodology" should be "3. Methodology", as "2" was already used in the prior section.

2.1 (line 210) - What is an example of an infectious disease that cannot be encountered in the naval military environment?

Page 9 - I might have spent at least a moment considering brucellosis, given that it is potentially weaponizable.

Page 10 - A great deal of the risk of severe cholera are due to host factors (e.g., malnutrition) that would be very rare among a military ship's crew.

Page 10 and 11 - You may be underestimating the morbidity of chikungunya and dengue, which could have a significant impact on a given sailor's ability to carry on their duties at sea. I agree that shipboard transmission is very unlikely, though.

Page 12 - "Enterohemorrhagic E coli" is better termed as "Shiga toxin-producing E coli (STEC)". I am compelled to mention that there have been sizable outbreaks of STEC in naval forces (specifically, the U.S. Marine Corps) in recent years (see Volk CG, et al. Crit Care Explor. 2021 May 18;3(5):e0423. doi: 10.1097/CCE.0000000000000423. - as an aside, I do not suggest that the authors cite this paper: It just indicated that a military STEC outbreak is not as mild or as rare as they are proposing). 

Page 14 - Hemolytic uremic syndrome is principally a sequela of EHEC/STEC (see above) and should be addressed with it.

Page 15 - I doubt very much that the number of human papillomavirus infections in France per year is "very low". As it is not particularly acute or prone to outbreaks, however, I would omit this disease from the list.

Page 15 - "Flu" is slang; use "influenza" consistently. 

Page 18 - Mentioning measles transmission without mention of vaccination seems odd to me.

Page 19 - I would specify that you mean meningococcal meningitis here, not just "meningitis".

Page 21 - Why is non-falciparum malaria mentioned separately from falicaprum malaria? I would just combine these both as "malaria".

Page 25 - Since the majority of seasonal influenza has been A/H1N1 for several years, I am uncertain why "seasonal influenza" is listed separately from A/H1N1.

Page 28 - Why is typhoid fever mentioned separately from salmonellosis?

Page 28 - Typhus is transmitted by arthopod vectors (e.g., fleas), not by touch.

Page 28 - Varicella section should include mention of vaccination.

Page 29-30 - Hepatitis A-D should be alphabetized under "H", not under "V". Again, uncertain why vaccination against HAV and HBV is not mentioned here.

Comments on the Quality of English Language

The quality of written English, although vastly superior to my French, would benefit from careful review by an experienced writer. There are minor, but relatively frequent, errors in word order and selection that can be revised to improve clarity and accuracy.

Author Response

Comment 1: Thank you for the opportunity to review this manuscript. This paper reviews epidemiologic risks of disease transmission onboard naval vessels. The paper includes an interesting discussion of modes of disease transmission. I would encourage inclusion in the discussion of more information regarding disease mitigation onboard ships, not just a list of diseases. The list of diseases is interesting but inconsistent in quality, mentions effective vaccines only on occasion, and makes some assumptions about risk that may not be accurate.

Answer 1: We completely agree that focusing on disease mitigation strategies onboard ships is essential. Beyond listing diseases, it is crucial to explore innovative solutions that can enhance detection, prevention, and elimination of pathogens in confined naval environments.

Better identification and understanding of pathogens will improve detection (e.g., PCR tests, air quality measurements), prevention (e.g., awareness of barrier gestures, isolation), and elimination (e.g., air or water decontamination). These measures can be integrated into ship design through technological advancements and cost-benefit analyses.

Currently, several advanced solutions are being studied for implementation on military vessels:

  • Enhanced Air Filtration Systems: High-Efficiency Particulate Air (HEPA) filters and Ultraviolet Germicidal Irradiation (UVGI) are being tested to reduce airborne transmission of pathogens. These systems can be incorporated into ventilation networks to provide continuous air purification.
  • Real-time Pathogen Detection: The development of onboard biosensors capable of detecting airborne or waterborne pathogens in real time is a key area of research. These sensors use molecular detection techniques such as CRISPR-based assays or rapid PCR diagnostics.
  • Autonomous Decontamination Technologies: Self-disinfecting surfaces, antimicrobial coatings, and automated UV-C robots are being considered to reduce contamination risks in high-contact areas.
  • Quarantine and Medical Isolation Units: Future ship designs may include dedicated isolation spaces equipped with negative pressure systems to contain infectious diseases more effectively.
  • Advanced Water Treatment Systems: Improved ballast water management and onboard water purification technologies, such as ozone and plasma-based disinfection, are being evaluated to prevent the spread of waterborne pathogens.
  • Predictive Analytics: health monitoring systems could analyze crew health data, detect early signs of disease outbreaks, and optimize response strategies through predictive modeling.

By integrating these technologies, military ships can significantly enhance their resilience against infectious disease outbreaks, ensuring operational readiness while protecting crew health.

The discussion has been expanded in this regard.

Comment 2: Under "Literature review", the authors indicated finding only 65 papers using their selected search terms. When I enter their search terms into PubMed alone, I find over 650,000 entries. I assume that there were more criteria used in their search than described.

Answer 2: In the 'Literature Review' section, we acknowledge that the initial search generated a large number of results. It has been modified. However, to ensure the relevance and quality of the sources, we applied several additional inclusion and exclusion criteria, such as restrictions on the publication date, linguistic limitations, relevance to naval or maritime environments, as well as the exclusion of duplicate or non-relevant studies. We also prioritized peer-reviewed articles and studies specifically focusing on pathogen transmission, epidemics, and biological resilience in maritime environments. Additional exclusion criteria have been included in the article.

Comment 3: Under 2.1.1, the authors write that "the greatest risk of pathogen transmission remains airborne transmission." In what way do they mean "the greatest risk"? Is this a reference to the severity of the diseases commonly transmitted through the airborne route, or is it a reference to the relative contagiousness? (Norovirus, while not a particularly dangerous disease in military populations, is widespread on ship, spreads rapidly, and carries a high risk for outbreaks.)

Answer 3: The term 'pathogen transmission' has been replaced with 'contagiousness'

Comment 4: Under 2.1.2, the authors describe R0 as a key predictor of contagiousness. I would argue that R0 predicts nothing; rather, it is a mathematical approximation of observed disease transmission. I would consider rephrasing that sentence to reflect that R0 is a way to quantify contagiousness, not predict it per se. (The authors' subsequent description of how R0 is derived is very well-written, though.)

Answer 4: The sentence has been modified. “Key predictor” has been replaced by “as a means of quantifying contagiousness”

Comment 5: There is an error in the numbering of sections; section "2. Methodology" should be "3. Methodology", as "2" was already used in the prior section.

Answer 5: The numbering of sections has been corrected.

Comment 6: 2.1 (line 210) - What is an example of an infectious disease that cannot be encountered in the naval military environment?

Answer 6: The sentence has been completed. Bilharzia is an example of infectious disease that cannot be encountered in the naval military environment: “Diseases that cannot be encountered in the naval military environments due to their mode of transmission were excluded from the study (for example: Bilharzia).”

Moreover, the title of Table I has been completed: “Diseases description and risk prioritization calculation. Diseases that are not encountered in the military naval environment are listed, but they have not been classified according to the risk prioritization index (the boxes have been grayed out). TBR: Terrorist Biological Risk.”

Comment 7: Page 9 - I might have spent at least a moment considering brucellosis, given that it is potentially weaponizable.

Answer 7: Yes, considering brucellosis as a potential bioterrorism weapon is indeed relevant. Brucellosis is particularly concerning due to its ability to be aerosolized, making it a viable candidate for airborne dissemination. This characteristic allows it to infect individuals over a wide area with relative ease. Additionally, brucellosis requires only a low infectious dose to cause illness, meaning that even small quantities of the pathogen can lead to widespread infection.

These factors, combined with the difficulty in diagnosing brucellosis early due to its nonspecific symptoms, make it a potential biothreat.

Comment 8: Page 10 - A great deal of the risk of severe cholera are due to host factors (e.g., malnutrition) that would be very rare among a military ship's crew.

Answer 8: Indeed, the clinical severity of cholera is closely linked to individual characteristics, such as the host's nutritional status and overall health. These factors significantly influence the outcome of the infection. Therefore, in the case of a military ship's crew, where such factors like malnutrition are less common, the risk of severe cholera may be lower. This justifies applying a slightly lower coefficient for severity (coefficient 4). The modification has been made in the table accordingly.

Comment 9: Page 10 and 11 - You may be underestimating the morbidity of chikungunya and dengue, which could have a significant impact on a given sailor's ability to carry on their duties at sea. I agree that shipboard transmission is very unlikely, though.

Answer 9: Indeed, morbidity has likely been underestimated. We propose a severity rating of 3, because these diseases are known to cause debilitating muscular and joint pain.

Comment 10: Page 12 - "Enterohemorrhagic E coli" is better termed as "Shiga toxin-producing E coli (STEC)". I am compelled to mention that there have been sizable outbreaks of STEC in naval forces (specifically, the U.S. Marine Corps) in recent years (see Volk CG, et al. Crit Care Explor. 2021 May 18;3(5):e0423. doi: 10.1097/CCE.0000000000000423. - as an aside, I do not suggest that the authors cite this paper: It just indicated that a military STEC outbreak is not as mild or as rare as they are proposing). 

Answer 10: "Enterohemorrhagic E. coli" was the term that had been used. We understood that it should rather be "Shiga toxin-producing E. coli (STEC). The term has been modified in the table."

Comment 11: Page 14 - Hemolytic uremic syndrome is principally a sequela of EHEC/STEC (see above) and should be addressed with it.

Answer 11: Hemolytic uremic syndrome is indeed a major complication of infections caused by EHEC/STEC. Given this close association and the fact that this syndrome primarily affects children, who are vulnerable due to their developing immune system, and elderly individuals, whose immunity is often weakened, this syndrome has been removed from Table IVa.

Comment 12: Page 15 - I doubt very much that the number of human papillomavirus infections in France per year is "very low". As it is not particularly acute or prone to outbreaks, however, I would omit this disease from the list.

Answer 12: The notion of « very low” has been removed.

Comment 13: Page 15 - "Flu" is slang; use "influenza" consistently. 

Answer 13: « Flu » has been replaced by ‘influenza”.

Comment 14: Page 18 - Mentioning measles transmission without mention of vaccination seems odd to me.

Answer 14: A moderation coefficient is indeed mentioned in the table for this disease. These moderation coefficients account for vaccination. They are described in the "Methodology" section.

Comment 15: Page 19 - I would specify that you mean meningococcal meningitis here, not just "meningitis".

Answer 15: Meningococcal meningitis is always bacterial, but not all bacterial meningitis cases are meningococcal.

Viral meningitis is not a major concern in the naval field. Bacterial meningitis, particularly meningococcal meningitis, is highly contagious and can lead to death within hours.

Comment 16: Page 21 - Why is non-falciparum malaria mentioned separately from falicaprum malaria? I would just combine these both as "malaria".

Answer 16: « Non-falciparum malaria » was combined with “falciparum malaria”.

Comment 17: Page 25 - Since the majority of seasonal influenza has been A/H1N1 for several years, I am uncertain why "seasonal influenza" is listed separately from A/H1N1.

Answer 17: Seasonal influenza has been removed since it is indeed caused by the H1N1 virus. The unexpected severity aboard military ships may be due to a particular virulence or an inadequate vaccine, whose effectiveness can vary from year to year.

Comment 18: Page 28 - Why is typhoid fever mentioned separately from salmonellosis?

Answer 18: Typhoid fever and salmonellosis are both caused by bacteria from the Salmonella genus, but they are distinct, and one is not a consequence of the other.

  • Typhoid fever: It is caused by Salmonella enterica serotype Typhi (or sometimes Paratyphi). This infection is systemic, affecting the whole body, and can be severe if not treated. It is primarily transmitted through water and food contaminated with fecal matter.
  • Salmonellosis: It is usually caused by other serotypes of Salmonella enterica (such as Salmonella Enteritidis or Salmonella Typhimurium). Salmonellosis can occur as a result of foodborne outbreaks (TIAC – Toxi-Infection Alimentaire Collective). These outbreaks happen when multiple people consume contaminated food or water, leading to gastrointestinal infections. Salmonella is one of the most common bacteria responsible for TIAC. It spreads through undercooked meat, eggs, dairy products, or contaminated water. It primarily causes acute gastroenteritis (diarrhea, vomiting, abdominal cramps), but it generally does not spread throughout the body like typhoid fever.

In summary, typhoid fever is not a consequence of salmonellosis, but they are caused by bacteria from the same genus and are transmitted in similar ways Contamination through feces or contaminated food).

Comment 19: Page 28 - Typhus is transmitted by arthopod vectors (e.g., fleas), not by touch.

Answer 19: Yes, that's true. Typhus is transmitted by arthropod vectors, primarily lice that are infected, but also sometimes by fleas or mites. The virus is typically transmitted when these insects bite a person and their feces contaminate the wound. Typhus is not transmitted by direct contact with an infected person. The correction has been made in the table.

Comment 20: Page 28 - Varicella section should include mention of vaccination.

Answer 20: In France, vaccination against varicella is not systematic for the general population, unlike in some countries such as the United States. However, it is recommended for certain at-risk groups.

The French National Authority for Health (HAS) recommends varicella vaccination for:

  1. Adolescents aged 12 to 18 who have never had varicella, as adult cases tend to be more severe.
  2. Women of childbearing age without a history of varicella, to prevent complications during pregnancy.
  3. Healthcare and childcare professionals who are not immune, as they are in contact with at-risk individuals.

Vaccination schedule:

  • Two doses are required, at least four weeks apart for individuals over 13 years old.
  • For children aged 12 months to 12 years, a single dose may be sufficient, but two doses are recommended for better protection.

Although the varicella vaccine is not currently included in the military vaccination schedule, this may change. As the chickenpox vaccine is effective, a coefficient of 0.2 has been added to the disease table. A bibliographic reference has also been included.

Comment 21: Page 29-30 - Hepatitis A-D should be alphabetized under "H", not under "V". Again, uncertain why vaccination against HAV and HBV is not mentioned here.

Answer 21: ”Viral hepatitis A/B/C/D/E” has been modified by “Hepatitis A/B/C/D/E viral” in the table. A moderation coefficient is indeed mentioned in the table for hepatitis A. These moderation coefficients account for vaccination. They are described in the "Methodology" section.

Although the hepatitis B virus (HBV) is present in most biological fluids of infected people and is highly contagious, its probability of occurrence in a military naval environment is zero. Indeed, sailors are vaccinated.These has been specified in the disease table.

Comment 22: The quality of written English, although vastly superior to my French, would benefit from careful review by an experienced writer. There are minor, but relatively frequent, errors in word order and selection that can be revised to improve clarity and accuracy.

Answer 22: The article has been reviewed by a second native English speaker to ensure its accuracy and clarity

Reviewer 3 Report

Comments and Suggestions for Authors

This scientific literature review examines shipboard diseases and pathogen characteristics to understand their spread and elimination, aiming to ensure the biological resilience of military ships and mission success. By analyzing the characteristics of pathogens and disease risks, researchers establish a risk priority index to provide guidance for naval vessels in responding to infectious disease threats. The article lists 58 diseases that may occur in the naval military environment and evaluates and classifies them according to clinical severity and infectivity. It will help to find pathogens as soon as possible, control the threat of infectious diseases, and avoid the large - scale outbreak of a COVID - 19 - like epidemic on ships, which would have a serious impact on military operations. At the same time, it provides direction for subsequent related research.Here are several questions and suggestions for this article.

1. The introduction to the previous research questions was too brief, and several representative studies should be detailed.

2. The PRISMA used in the research strategy did not explain why this method was used for the study, nor did it explain the advantages of using this research method.

3. Due to the particularity of the research environment, it is necessary to analyze why the basic infectivity number R0 is used as a key factor and what limitations R0 has compared to other environments in the current environment.

4. Some specific professional terms in the article can be further explained.

5.The article proposes two modeling methods: “In the first approach, it can be modeled from retrospective analysis of outbreaks.” and “In the second approach, the quanta emission rate is modeled from the predictive estimate of the infectious particle load expelled by patient zero.” However, these two methods were not introduced.

6. The evaluation of clinical severity and infectious disease level proposed in the article is too subjective and does not reflect the process of obtaining this evaluation standard.

7. Chart 1 is too long to look good in the paper. It can be simply stored as an attachment and a more detailed analysis of the diseases mentioned in it can be provided.

8. The article discusses the sealing and confinement situations in other environments, such as hospitals, detention centers, airplanes, etc., and mentions their commonalities, but does not reflect any special features of ships in this closed environment compared to other environments.

9. Some of the references introduced in the article were not thoroughly analyzed and discussed.

10. The future research direction and plan are too vague, and should be supplemented to better guide subsequent research.

Author Response

Comment 1: The introduction to the previous research questions was too brief, and several representative studies should be detailed.

Answer 1: To strengthen the introduction to the research questions, several representative studies on infectious disease outbreaks in naval settings have been detailed. Historical outbreaks illustrate the unique challenges posed by confined environments such as ships, where close contact, shared ventilation systems, and prolonged interactions facilitate pathogen transmission:

“The study by De Laval et al. [9] showed that a large proportion of the crew of the French aircraft carrier Charles de Gaulle was infected, highlighting the rapid spread of SARS-CoV-2 in a confined environment. Crowded conditions, shared ventilation, and frequent interactions among sailors facilitated the transmission of the virus. The majority of cases were symptomatic, but most presented mild to moderate forms, with only a limited number of individuals requiring hospitalization. Nevertheless, despite attempts at isolation and social distancing on board, the virus spread before strict measures could be implemented, leading to the mission being cut short. On the USS Theodore Roosevelt aircraft carrier, more than 1,200 sailors were infected, representing approximately 25% of the crew. A portion of the crew was quarantined in Guam, which had a major impact on the ship's operational readiness. »

Comment 2: The PRISMA used in the research strategy did not explain why this method was used for the study, nor did it explain the advantages of using this research method.

Answer 2: The PRISMA methodology was chosen for this study to ensure a rigorous, and reproducible research process. PRISMA provides a structured approach to systematically identifying, selecting, and analyzing relevant literature, which is essential for synthesizing high-quality evidence on pathogen transmission in naval environments.

The use of PRISMA offers several advantages. First, it minimizes selection bias by following a predefined inclusion and exclusion criteria framework, ensuring that only relevant and high-quality studies are considered. Second, it enhances transparency by providing a detailed flow diagram of the study selection process, allowing for better reproducibility of the research. Third, PRISMA improves the reliability of findings by promoting a comprehensive and structured literature review process, facilitating the identification of key trends and gaps in existing knowledge.

By applying PRISMA, this study ensures methodological rigor while systematically assessing the impact of infectious diseases in naval settings, ultimately strengthening the validity of its conclusions and recommendations.

The section 2.1 “Search strategy for characterization of pathogens” has been completed.

Comment 3: Due to the particularity of the research environment, it is necessary to analyze why the basic infectivity number R0 is used as a key factor and what limitations R0 has compared to other environments in the current environment.

Answer 3: The sentence has been nuanced: “Key predictor” has been replaced by “as a means of quantifying contagiousness”. Although R0 presents some limits:

  • Influence of Ship-Specific Environmental Factors
  • Ship ventilation systems can alter the spread of airborne pathogens, affecting transmission dynamics compared to terrestrial environments.
  • The high crew density and confined spaces increase close contacts, making R₀ values estimated from other environments less applicable.
  • Exclusion of Control Measures
  • R₀ assumes a fully susceptible population, which is not always the case on board due to strict health protocols and vaccination programs.
  • Pathogens with a high R₀ can be controlled by effective measures, making the raw R₀ value less meaningful in practice.
  • Contextual Variability and Estimation Challenges
  • R₀ estimates vary based on the initial conditions of each outbreak (population size, pre-existing immunity, etc.).
  • On long missions, the risk may progressively increase due to crew fatigue and limited access to medical care.

R0 is a key factor because:

  • R₀ helps evaluate the potential spread of an infectious disease within a specific environment, such as a military vessel.
  • Identifying pathogens with a high R₀, it is possible to prioritize infection prevention and control measures on board.
  • R₀ is a critical parameter in epidemiological models used to estimate the effectiveness of containment, ventilation, or vaccination strategies in enclosed environments.

Section 2.2.2 has been completed.

Comment 4: Some specific professional terms in the article can be further explained.

Answer 4: Most of the specific terms have been defined:

  • PRISMA (Preferred Reporting Items for Systematic Reviews and Meta-Analysis): The PRISMA methodology was chosen for this study to ensure a rigorous, and reproducible research process. PRISMA provides a structured approach to systematically identifying, selecting, and analyzing relevant literature, which is essential for synthesizing high-quality evidence on pathogen transmission in naval environments. The use of PRISMA offers several advantages. First, it minimizes selection bias by following a predefined inclusion and exclusion criteria framework, ensuring that only relevant and high-quality studies are considered. Second, it enhances transparency by providing a detailed flow diagram of the study selection process, allowing for better reproducibility of the research. Third, PRISMA improves the reliability of findings by promoting a comprehensive and structured literature review process, facilitating the identification of key trends and gaps in existing knowledge.
  • R0 (Basic Reproduction Number): R0 is defined as the average number of secondary infectious produced by an infected individual in a susceptible host population. When R0 is less than 1, each infected individual, on average, transmits the pathogen to fewer than one other individual, leading to the pathogen’s eventual disappearance from the population. Conversely, when R0 is greater than 1, the number of cases increases on average over time, potentially leading to an epidemic [19]. Huang et al., (2021) [20] demonstrated in their study that a higher asymptomatic ratio leads to more infectious contacts.
  • TBR (Terrorist Biological Risk): Several agents can be intentionally used to infect people, leading to a significant number of illnesses and deaths. This act is known as bioterrorism
  • Biological resilience: Biological resilience can be defined as “the ability to prevent, limit the effects of, and bounce back from biological disturbance linked to pathogenic elements of natural or terrorist origin. The aim is to enable military ships to recover function and operational efficiency as effectively as possible to a state equivalent to the prior disturbance” [12–14]. This biological resilience is based on a systemic approach to natural or terrorist biological risks, establishing hypotheses to ensure the operational effectiveness of ships.
  • Quanta per hour: The emission factor is measured in quanta per hour (quanta.h-1). A quantum corresponds to the dose of aerosol sufficient to infect 63% of susceptible individuals. Emission factors depend on the symptomatic nature of the infection, physical activity, and vocalization.

Comment 5: The article proposes two modeling methods: “In the first approach, it can be modeled from retrospective analysis of outbreaks.” and “In the second approach, the quanta emission rate is modeled from the predictive estimate of the infectious particle load expelled by patient zero.” However, these two methods were not introduced.

Answer 5: Initially, this study aimed to monitor epidemics that had occurred aboard military ships. This surveillance of epidemics, along with the COVID-19 epidemic, showed that outbreaks aboard military vessels could endanger the success of their missions. In a second phase, the goal was to make our ships resilient by limiting the impact of epidemics on their missions. This is why we developed a rating system based on severity and contagiousness, in order to create a risk prioritization index and rank diseases relative to each other. This ranking highlights the diseases with the highest likelihood of occurrence aboard our military ships. The current goal is to find innovative solutions to reduce the risk of epidemics on board, building upon previous work, particularly the probability of disease occurrence.

Comment 6: The evaluation of clinical severity and infectious disease level proposed in the article is too subjective and does not reflect the process of obtaining this evaluation standard.

Answer 6: This point has been added to the limitations of the study to acknowledge the subjectivity of the evaluation of clinical severity and infectious disease levels, as well as the lack of a standardized process for obtaining these assessments: “The subjectivity of the evaluation of clinical severity and contagiousness, although based on scientific publications, is also a limitation of this study”.

Comment 7: Chart 1 is too long to look good in the paper. It can be simply stored as an attachment and a more detailed analysis of the diseases mentioned in it can be provided.

Answer 7: The table has been divided into several sub-tables, separating parasites, viruses, bacteria, and fungi for easier reading.

A comparison of the diseases with the highest probability of occurrence in the military naval environment has been developed:

“The most high-risk diseases in a military naval environment share several characteristics in terms of contagiousness and severity. They can be categorized into three main modes of transmission: direct contact with bodily fluids (Ebola, rabies, smallpox, hepatitis E), fecal-oral transmission through contaminated water or food (cholera, Shiga toxin-producing bacteria, shigellosis, hepatitis E), and airborne or droplet transmission (meningitis, pneumonic plague, tuberculosis, smallpox). Among them, smallpox, pneumonic plague, and tuberculosis are particularly dangerous due to their airborne transmission in confined spaces, facilitating rapid contagion aboard a ship. Similarly, fecal-oral diseases such as cholera and shigellosis pose a major risk in an environment where potable water and food supplies are centralized, increasing the potential for outbreaks. Some diseases, such as Ebola and rabies, are less contagious but extremely lethal. Finally, tuberculosis and hepatitis E can develop into chronic infections, posing a persistent risk to crews. In a military naval setting, where close quarters and limited resources make it difficult to isolate infected individuals and implement rapid countermeasures, these diseases represent a significant threat to operational health and readiness.”

Comment 8: The article discusses the sealing and confinement situations in other environments, such as hospitals, detention centers, airplanes, etc., and mentions their commonalities, but does not reflect any special features of ships in this closed environment compared to other environments.

Answer 8: Unlike hospitals, where medical infrastructure is readily available, or airplanes, where exposure time is limited, ships operate in prolonged isolation with limited medical resources and evacuation options. Additionally, the constant movement of the vessel, variations in ventilation efficiency, and shared living quarters create a unique epidemiological context. We will expand the discussion to better emphasize these specific characteristics and their implications for disease transmission and containment strategies.

Comment 9: Some of the references introduced in the article were not thoroughly analyzed and discussed.

Answer 9: Some of the references have been further discussed. For example, in the introduction the references [9] and [10] have been developed: “The study by De Laval et al. [9] showed that a large proportion of the crew of the French aircraft carrier Charles de Gaulle was infected, highlighting the rapid spread of SARS-CoV-2 in a confined environment. Crowded conditions, shared ventilation, and frequent interactions among sailors facilitated the transmission of the virus. The majority of cases were symptomatic, but most presented mild to moderate forms, with only a limited number of individuals requiring hospitalization. Nevertheless, despite attempts at isolation and social distancing on board, the virus spread before strict measures could be implemented, leading to the mission being cut short. On the USS Theodore Roosevelt aircraft carrier, more than 1,200 sailors were infected, representing approximately 25% of the crew. A portion of the crew was quarantined in Guam, which had a major impact on the ship's operational readiness [10]”

Comment 10: The future research direction and plan are too vague, and should be supplemented to better guide subsequent research.

Answer 10: The future research will focus on identifying the common characteristics of the pathogens that pose the greatest risk to military operations. These characteristics will be used to design targeted strategies for pathogen control. Specifically, the aim is to identify key factors such as resilience to environmental conditions, modes of transmission, and susceptibility to available countermeasures. This will allow for the prioritization of pathogens that require immediate intervention.

Future studies will incorporate experimental trials conducted in controlled environments that simulate maritime conditions. These trials will evaluate the effectiveness of various pathogen detection, identification, containment, and elimination methods, taking into account factors such as humidity, salinity, and vibrations, which are specific to maritime settings.

Numerical simulations, particularly Computational Fluid Dynamics, will be used to model how pathogens spread within the confines of a ship. This will enable the assessment of prevention strategies and the optimization of pathogen detection systems, which are crucial in mitigating the spread of infections in closed environments.

 Given the high risk of airborne transmission identified in the current study, future research will prioritize investigating this mode of transmission. Specifically, the research will focus on understanding airborne pathogen dynamics within the unique air circulation systems aboard military ships, assessing the impact of air filtration systems, and exploring the effectiveness of interventions such as air purifiers and ventilation adjustments.

Future studies will aim to analyze the daily activities of sailors to identify specific behaviors and times of day that facilitate disease transmission. This analysis, while challenging due to access restrictions on military vessels, will be supported by agent-based simulation models. These models will replicate sailors' behaviors and environmental factors, allowing for the identification of high-risk activities and the development of targeted strategies to reduce transmission.

The discussion has been expanded based on these elements.

Round 2

Reviewer 2 Report

Comments and Suggestions for Authors

Thank you for the opportunity to review this revised manuscript. Overall, the draft is improved, and I appreciate the authors' responses. My remaining observations are as follows:

Lines 93-106 - PRISMA is a standardized system; I do not think it is necessary to describe what it is in detail or explain its advantages.

Line 128 - "Fomite" does not need to be in quotation marks.

Line 345 - I would not use "fevers" as a synonym for "infections".

Table 4

  • should say "hepatitis A virus", not "viral" (same for B-E)
  • mention hepatitis A vaccination
  • I am not sure that HIV is "always highly contagious"
  • Would remove HPV entirely
  • H1N1 influenza is not particularly limited, especially this year, and influenza outbreaks on warships are not rare
  • Marburg is misspelled
  • Measles vaccination should be mentioned; we don't have measles outbreaks on warships for a reason
  • Meningitis is a syndrome, not a specific disease - please describe particular pathogens, not "meningitis" collectively
  • On table IVd, "fungus" is misspelled as "fungis"

Author Response

Comment 1: Lines 93-106 - PRISMA is a standardized system; I do not think it is necessary to describe what it is in detail or explain its advantages.

Answer 1: In response to the requests from the other two reviewers, we were asked to clarify and elaborate the definition and advantages of PRISMA. To address their concerns, we provided additional details while ensuring that the description remains concise and relevant to the study. We hope this strikes a balance between providing necessary context and avoiding excessive explanation.

Comment 2: Line 128 - "Fomite" does not need to be in quotation marks.

Answer 2: The quotation marks were initially used to emphasize the term, but we can remove them to align with standard formatting. We make the necessary correction.

Comment 3: Line 345 - I would not use "fevers" as a synonym for "infections".

Answer 3: We have replaced the word 'fever' with 'infections'.

Comment 4: Table 4 - should say "hepatitis A virus", not "viral" (same for B-E)

Answer 4: We will revise the text to replace "viral" with "virus" for hepatitis A, as well as for types B to E

Comment 5: Table 4 - mention hepatitis A vaccination

Answer 5: The hepatitis A vaccination is mentioned in Table IV because a coefficient of 0.2 has been added. Furthermore, Reference 41 has been added, line 306. The sentence “The disease is preventable with vaccine” has been added.

Comment 6: Table 4 - I am not sure that HIV is "always highly contagious"

Answer 6: You're correct to question the statement that HIV is "always highly contagious." The contagiousness of HIV can vary significantly depending on several factors. Here are some key points to consider:

  • Viral load: the amount of HIV in the blood, known as the viral load, is a crucial factor in determining contagiousness. Individuals with a high viral load are more likely to transmit the virus than those with a low or undetectable viral load. Effective antiretroviral therapy can reduce the viral load to undetectable levels, significantly reducing the risk of transmission.
  • Mode of transmission: HIV is primarily transmitted through certain body fluids, such as blood, semen, vaginal fluids, and breast milk. The risk of transmission varies depending on the type of exposure.
  • Use of prevention methods: The use of condoms, pre-exposure prophylaxis, and post-exposure prophylaxis can significantly reduce the risk of HIV transmission. These methods, along with regular testing and treatment, are essential components of HIV prevention strategies.

 In summary, HIV is not always highly contagious. The risk of transmission depends on factors such as viral load, mode of transmission, and the use of prevention methods. Effective treatment and prevention strategies can significantly reduce the contagiousness of HIV.

The sentence “always highly contagious” has been deleted.

Comment 7: Table 4 - Would remove HPV entirely

Answer 7: HPV has been removed from the table because the occurrence of this disease is unlikely onboard military vessels.

Comment 8: Table 4 - H1N1 influenza is not particularly limited, especially this year, and influenza outbreaks on warships are not rare

Answer 8: The sentence has been modified “This disease was widespread during the 2009 epidemic. The disease has a short incubation period and moderate contagiousness and it is preventable with the seasonal influenza vaccine whose effectiveness may vary depending on the years.”

Comment 9: Table 4 - Marburg is misspelled

Answer 9: The spelling has been corrected.

Comment 10: Table 4 - Measles vaccination should be mentioned; we don't have measles outbreaks on warships for a reason

Answer 10: It is correct to consider meningitis as a syndrome, as it has multiple causes and is not a single, specific disease. However, it is important not to confuse it with meningeal syndrome, which refers to the set of clinical signs and symptoms observed in meningitis. Meningitis is characterized by inflammation of the meninges, the protective membranes covering the brain and spinal cord. It can be caused by a variety of pathogens, including bacteria, viruses, fungi, and parasites. Here are some specific pathogens known to cause meningitis:

  • Bacterial meningitis: Neisseria meningitidis, Haemophilus influenzae, Streptococcus pneumoniae, Listeria monocytogenes
  • Viral meningitis: Enteroviruses, herpes simplex virus, varicella-zoster virus, mumps virus
  • Fungal meningitis: Cryptococcus neoformans, Histoplasma capsulatum

Not all cases of meningitis are infectious (e.g., carcinomatous meningitis). Among bacterial meningitis, transmission can occur through respiratory routes (e.g., Neisseria meningitidis, Streptococcus pneumoniae, Haemophilus influenzae) or foodborne routes (e.g., Listeria monocytogenes).

For viral meningitis, herpetic meningoencephalitis can be severe but usually remains isolated. Other viral causes include poliomyelitis and enterovirus infections, which are transmitted through the fecal-oral route; however, vaccination is available for poliovirus.

The names of pathogens have been added to the table for each type of meningitis.

Comment 11: Table 4 - On table IVd, "fungus" is misspelled as "fungis"

Answer 11: The spelling has been corrected.